# Towards Fast Graph Generation via Autoregressive Filtration Modeling

## Abstract

Graph generative models often face a critical trade-off between learning complex distributions and achieving fast generation speed. We introduce Autoregressive Filtration Modeling (AFM), a novel approach that addresses both challenges. AFM leverages filtration, a concept from topological data analysis, to transform graphs into short sequences of monotonically increasing subgraphs. This enables a structured autoregressive generation process, contrasting with the stochastic trajectories of diffusion models. We propose a novel autoregressive graph mixer model to learn this filtration process, coupled with a noise augmentation strategy to mitigate exposure bias and a reinforcement learning approach to refine the generative model. Extensive experiments on diverse synthetic and real-world datasets demonstrate AFM's superior performance compared to existing autoregressive models. Additionally, AFM achieves a 100-fold speedup in generation time compared to state-of-the-art diffusion models while maintaining the quality of generated graphs. This work represents a significant advancement towards high-throughput graph generation.

## 1 Introduction

Graphs are fundamental structures that model relationships in various domains, from social networks and molecular structures to transportation systems and neural networks. The ability to generate realistic and diverse graphs is crucial in many applications, such as drug discovery (Liu et al., 2018; Vignac et al., 2023), network simulation (Yu & Gu, 2019), and protein design (Ingraham et al., 2019). The space of drug-like molecules and protein conformations is, for practical purposes, infinite, limiting the effectiveness of in-silico screening of existing libraries (Polishchuk et al., 2013; Levinthal, 1969). Consequently, high-throughput graph generation—the task of efficiently creating new graphs that faithfully emulate properties similar to those observed in a given domain—has thus emerged as a critical challenge in machine learning and generative artificial intelligence (Gangwal et al., 2024; Grisoni et al., 2021).

While significant progress has been made in graph generation, existing approaches often face expressive or computational challenges. Classical methods, such as Erdős-Rényi models (Erdos et al., 1960) or stochastic block models (Holland et al., 1983), are typically tailored to model specific graph families, thus struggling to capture the heterogeneous structural properties of real-world graphs (You et al., 2018b). More recent deep learning-based approaches, particularly autoregressive (You et al., 2018b; Liao et al., 2019; Kong et al., 2023) and diffusion models (Vignac et al., 2023; Bergmeister et al., 2024), have shown promise in generating increasingly realistic graphs. However, these methods often suffer from scalability issues when handling large graphs, primarily due to their quadratic or higher computational complexity with respect to the number of nodes. Furthermore, many current diffusion-based approaches rely on iterative refinement processes involving a large number of steps. This computational burden not only limits the applicability of these models to large-scale graphs but also hinders their potential for high-throughput applications (Gentile et al., 2022; Gómez-Bombarelli et al., 2016; Polishchuk et al., 2013).

Recent work has explored the use of topological data analysis, particularly persistent homology and filtration (Edelsbrunner et al., 2002; Zomorodian & Carlsson, 2005), for graph representation. A filtration provides a multi-scale view of a given graph structure by constructing a nested sequence of subgraphs. This approach has shown promise in various graph analysis tasks, including classifi-

cation and similarity measurement (O'Bray et al., 2021; Schulz et al., 2022). In the context of generative modeling, filtration-based representations have been used to develop more expressive tools for generative model evaluation (Southern et al., 2023). However, the application of filtration-based methods for graph generation remains unexplored.

In this paper, we introduce Autoregressive Filtration Modeling (AFM), a novel approach to fast graph generation that models filtration sequences autoregressively. To generate a graph, our method produces a short sequence of increasingly dense and detailed subgraphs. Compared to diffusion models (Vignac et al., 2023; Bergmeister et al., 2024), AFM requires fewer iterations during sampling, resulting in significantly faster inference speed. Moreover, AFM incorporates advanced techniques to mitigate exposure bias (Bengio et al., 2015), a common challenge in existing autoregressive models, thereby improving sample quality. Our method offers a promising balance between efficiency and accuracy in graph generation, addressing key limitations of current approaches.

In summary, our contributions are as follows:

- We propose a novel autoregressive graph generation framework that leverages graph filtration to enable fast, high-throughput sampling.
- We introduce a specialized autoregressive model architecture designed to learn the unique properties of filtration sequences, incorporating both structural and temporal information about filtration.
- We identify exposure bias as a potential challenge in autoregressive graph generation and propose noise augmentation and adversarial fine-tuning as effective strategies to mitigate this issue.
- We conduct comprehensive ablation studies to evaluate the impact of different components within our framework, demonstrating that noise augmentation and adversarial fine-tuning substantially improve performance.
- Our empirical results highlight the strong performance and efficiency of our model compared to recent baselines. Notably, our model achieves inference speed 100 times faster than existing diffusion-based models.

## 2 RELATED WORK

AFM builds on the concept of graph filtration and is fine-tuned via reinforcement learning. In the following, we provide a brief overview of related graph generative models, training schemes, and applications of graph filtrations.

**Graph Generation.** Historically, simple statistical models (Erdos et al., 1960; Holland et al., 1983) have been used to model distributions of graphs. While these approaches lend themselves to theoretical investigation, they are typically insufficient to model datasets seen in real-world applications. GraphRNN (You et al., 2018b) made advances towards deep generative graph models by autoregressively generating nodes and their incident edges to build up an adjacency matrix. In a similar fashion, DeepGMG (Li et al., 2018) iteratively builds a graph node-by-node. Liao et al. (2019) proposed a more efficient autoregressive model by generating multiple nodes at a time in a block-wise fashion, leveraging mixtures of multivariate Bernoulli distributions in a similar way to us. These models share the fact that they build graphs via node-addition and therefore require some choice of node ordering, analogous to the choice of edge ordering that we make in our work via a filtration function. To avoid an arbitrary choice, OM (Chen et al., 2021) and GraphArm (Kong et al., 2023) learn node orderings, where the latter uses reinforcement learning to jointly learn the ordering policy alongside the generative model. In contrast to autoregressive node-addition methods, approaches by Goyal et al. (2020) and Bacciu et al. (2020) generate graphs through edge-addition following a pre-defined edge ordering. While this strategy bears similarities to our proposed filtration method, our approach distinctly differs by modeling sequences of graphs and allowing for edge deletion as well as addition. Graph variational autoencoders (Kipf & Welling, 2016; Simonovsky & Komodakis, 2018) generate all edges at one time, thereby reducing computational costs during inference. However, these methods struggle to model complicated distributions and may fail in the presence of isomorphic nodes (Zhang et al., 2021). Generative adversarial networks (GANs) (Bojchevski et al., 2018; Cao & Kipf, 2018; Martinkus et al., 2022) are likelihood-free and avoid the node-orderings and graph matching algorithms required in autoregressive models and VAEs. Graph diffusion models such as EDP-GNN (Niu et al., 2020) and GDSS (Jo et al., 2022), based on score

matching, or DiGress (Vignac et al., 2023), based on discrete denoising diffusion (Austin et al., 2021), have emerged as powerful generators. Unfortunately, they require many iterative denoising steps, making them slow during sampling. Some efforts have been made to increase inference speed. Improvements were achieved in autoregressive models by generating multiple graph features in a single step (Liao et al., 2019; Kong et al., 2023) while hierarchical approaches allowed scaling diffusion models (Bergmeister et al., 2024) and autoregressive models (Karami, 2024) to larger graphs. Additional efficiency gains have been achieved in diffusion models by using absorbing state processes (Chen et al., 2023).

**Reinforcement Learning Finetuning.** Ranzato et al. (2016) demonstrated that reinforcement learning (RL) allows optimizing non-differentiable sequence-level metrics in autoregressive modeling and argued that training in free-running mode mitigates exposure bias (Bengio et al., 2015). SeqGAN (Yu et al., 2017), which is most relevant to our work, avoids any extrinsic metrics by using a discriminator to provide feedback to the generative model. In the context of graph learning, You et al. (2018a) train a generative model for molecules via reinforcement learning, combining adversarial and domain-specific rewards. In contrast to our work, they do not consider general graphs and do not use any autoregressive pre-training. Taiga (Mazuz et al., 2023) uses reinforcement learning to optimize chemical properties of molecules obtained from a language model that is pre-trained on SMILES strings. Even graph diffusion models have been shown to be amenable to RL finetuning, allowing extrinsic non-differentiable metrics to be optimized (Liu et al., 2024).

**Graph Filtration.** Filtration is commonly used in the field of persistent homology (Edelsbrunner et al., 2002) to extract features of geometric data structures at different resolutions. Previously, graph filtration has mostly been used to construct graph kernels (Zhao & Wang, 2019; Schulz et al., 2022) or extract graph representations that can be leveraged in downstream tasks such as classification (O'Bray et al., 2021). While filtration has also been used for evaluating graph generative models (Southern et al., 2023), *to the best of our knowledge, our work presents the first model that directly leverages filtration for generation.*

## 3 METHOD

In this section, we present the Autoregressive Filtration Modeling (AFM) approach for graph generation. We begin with the notion of filtration and present various filtration strategies in Sec 3.1. Then, we introduce our autoregressive model in Sec. 3.2. Finally, in Sec. 3.3, we propose a two-staged training scheme for AFM.

In the following, we consider unlabeled and undirected graphs, denoted by $G = (V, E)$, where $V$ is the set of vertices and $E \subseteq V \times V$ is the set of edges. Without loss of generality, we assume $V = \{1, 2, \ldots, n\}$ and denote by $e_{ij}$ the edge between nodes $i, j \in V$. We assume that only connected graphs are presented to our model during training and filter training sets if necessary. Our approach is fundamentally based on the concept of graph filtration.

### 3.1 GRAPH FILTRATION

A filtration of a graph $G$ is defined as a nested sequence of subgraphs:

$$G = G_T \supseteq G_{T-1} \supseteq \cdots \supseteq G_1 \supseteq G_0 = (V, \emptyset) \tag{1}$$

where each $G_t = (V, E_t)$ is a graph sharing the same node set as $G_T := G$. The filtration satisfies the following properties: (1) $E_t \subseteq E_{t'}$ for all $t < t'$ and (2) $G_0$ is the completely disconnected graph, *i.e.,* $E_0 = \emptyset$. In our experiments, we choose $T = 15$ or $T = 30$, depending on the dataset.

A convenient method to define a filtration of $G$ involves specifying two key components (O'Bray et al., 2021): a filtration function defined on the edge set $f : E \to \mathbb{R}$ and a non-decreasing sequence of scalars $(a_0, a_1, \ldots, a_T)$ with $-\infty = a_0 \le a_1 \le \cdots \le a_{T-1} \le a_T = +\infty$. Given these components, we can define the edge sets $E_i$ as nested sub-levels of the function $f$:

$$E_t := f^{-1}((-\infty, a_t]) = \{e \in E \ : \ f(e) \le a_t\} \qquad \forall \, t = 1, \ldots, T-1. \tag{2}$$

The sequence $(a_t)_{t=0}^T$ is referred to as the *filtration schedule sequence*. The choice of the filtration function and the schedule sequence plays a crucial role in the effectiveness of graph filtration for generation. We present a visual example of the filtration process in Figure 1a. In the following, we will discuss several strategies for the filtration function and schedule.

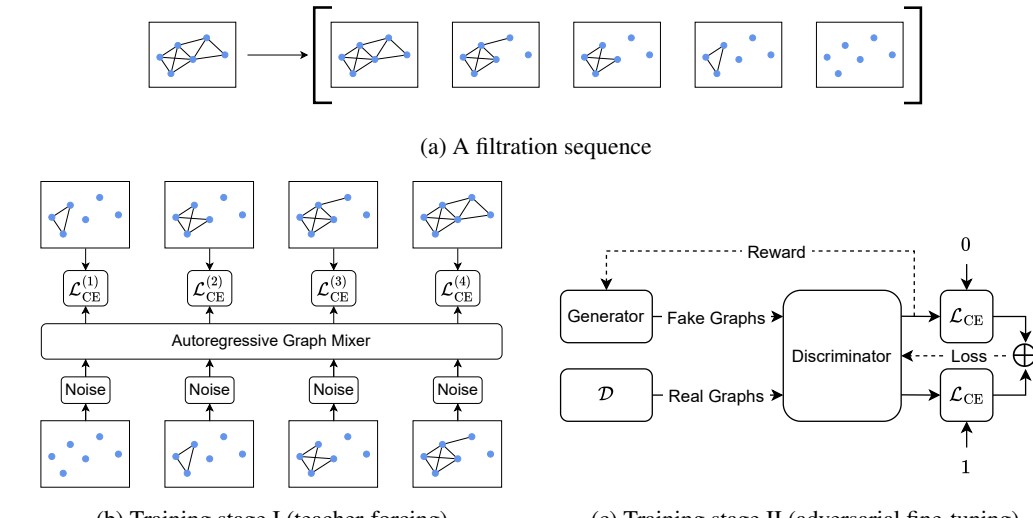

(a) A filtration sequence

(b) Training stage I (teacher-forcing)   (c) Training stage II (adversarial fine-tuning)

Figure 1: Top: A graph is transformed into a sequence of subgraphs (filtration) via edge-deletion. Bottom left: the generator is trained via teacher-forcing to reverse the filtration process. Bottom right: the generator is fine-tuned in free-running mode via reinforcement learning based on a reward signal output by a discriminator in a SeqGAN-like framework (c.f. details in Appendix A.18).

**Filtration Function.** In principle, any real-valued function defined on the set of node pairs can be used as the filtration function. For graph generation, an effective filtration function should satisfy two key criteria: *diversity* and *structural consistency*. Specifically, diversity means that the function should assign distinguishable values to different edges, facilitating a more granular filtration sequence. Structural consistency describes that edges with similar structural properties or proximity in the graph should receive similar weights from the filtration function. Based on these criteria, we propose the following filtration functions:

- *Line Fiedler function $f_{Fiedler}$:* This function is derived from the second smallest eigenvector (Fiedler vector) of the symmetrically normalized Laplacian of the line graph $L(G)$. In $L(G)$, nodes represent edges in the original graph $G$, and two nodes are connected if their corresponding edges in $G$ share a common vertex. This vector is not unique due to multiplicities of eigenvalues and sign flip preservation (i.e., its negative is also an eigenvector with the same eigenvalue). We thus choose it arbitrarily with a random sign choice per graph. This function captures global structural information and tends to assign similar values to neighboring edges.
- *Centrality functions $f_{between}$ and $f_{remote}$:* Following (Anthonisse, 1971; Brandes, 2008), we utilize the concept of betweenness-centrality to quantify the importance of an edge in facilitating communication between different parts of the graph. We refer to Appendix A.16 for details.

Our empirical comparison (detailed in Appendix A.16) demonstrates that the line Fiedler function generally outperforms the other two options. Consequently, we primarily utilize this function in our experiments. By employing our proposed filtration function, we can effectively guide the graph generation process. While we focus on pre-determined filtration functions, it is worth noting that learning edge weights dynamically, in a similar fashion as in Kong et al. (2023), presents an intriguing avenue for future research.

**Filtration Schedule Sequence.** Analogous to noise schedules in diffusion models (Nichol & Dhariwal, 2021), our filtration schedule governs the rate at which edges are added during sampling or removed during filtration. We model this process using a continuous scheduling function $\gamma : [0,1] \to [0,1]$ where $\gamma(0) = 0$ and $\gamma(1) = 1$. Given a graph, the discretized schedule sequence $(a_t)_{t=0}^T$ can then be defined as follows:

$$a_t := \inf \left\{ a \in \mathbb{R} \,:\, |f^{-1}((-\infty, a_t])| \geq |E| \cdot \gamma(t/T) \right\} \qquad \forall\, 0 \leq t \leq T, \qquad (3)$$

where $f$ is the filtration function and $T$ is the total number of steps. *We note that the discretized schedule $(a_t)_{t=0}^{T}$ is defined per graph, as it depends on $f$ and the cardinality $|E|$.*

Drawing inspiration from noise scheduling in diffusion models, we propose three scheduling functions:

- *Linear schedule $\gamma(t) := t$:* This schedule intends to add approximately the same of number of edges in each step. Assuming mostly distinct edge weights, this results in a roughly linear increase in the density of graphs $G_t$ over time.
- *Convex schedule $\gamma(t) := 1 - \cos(\pi t/2)$:* This schedule adds more edges in later steps, potentially offering finer control over the graph generation process in its earlier stages.
- *Concave schedule $\gamma(t) := \sin(\pi t/2)$:* This schedule adds more edges during the initial steps, which may help establish the overall graph structure more quickly while leaving room for finer adjustments over structural details in the later stages.

We conducted an empirical comparison of these schedules on a planar graph dataset in Appendix A.16. Our findings indicate that no single variant consistently outperforms the others across all evaluation metrics. However, the concave schedule achieves the highest validity score. In the experiments we present in the main paper, we use the linear schedule for simplicity.

## 3.2 AUTOREGRESSIVE MODELING OF THE FILTRATION SEQUENCE

Our objective is to develop a generative model that reverses the graph filtration process in Eqn. (1). Given a node set $V$, we aim to generate a sequence of graphs $G_0, G_1, \ldots, G_T$ on $V$, where each subsequent graph contains an increasing number of edges. The final graph $G_T$ should plausibly represent a sample from the target data distribution. We formulate this generative process using an autoregressive model, expressing the joint likelihood as follows:

$$p_\theta(G_T, \ldots, G_0) = p(G_0) \prod_{t=1}^{T} p_\theta(G_t | G_{t-1}, \ldots, G_0), \tag{4}$$

where $p(G_0)$ represents the initial distribution, which we define as a point mass on the fully disconnected graph $(V, \emptyset)$. In the following sections, we will detail our implementation of the autoregressive model $p_\theta$, including the architecture and training procedure. While existing autoregressive models typically utilize RNNs (You et al., 2018b; Liao et al., 2019; Goyal et al., 2020; Bacciu et al., 2020) or a first-order autoregressive structure (Kong et al., 2023), our model architecture for implementing $p_\theta$ is a novel and efficient design inspired by MLP-Mixers (Tolstikhin et al., 2021).

**Backbone Architecture.** The filtration sequence can be viewed as a dynamic graph with a constant node set but evolving edge sets. Our backbone architecture operates on this structure by alternating between two types of information processing layers. The first type, called structural mixing, consists of a GNN that processes graph structures $G_0, \ldots, G_{t-1}$ independently, with weights shared across time steps. The second type, called temporal mixing, consists of a transformer decoder that processes node representations along the temporal axis, with weights shared across nodes. Formally, given input node representations $v_i^{(t)} \in \mathbb{R}^D$ for nodes $i \in V$ and time steps $t \in [T-1]$, a single mixing operation in our backbone model produces new representations $\hat{v}_i^{(t)}$ and is defined as:

Structural mixing: $\quad \left(\tilde{v}_i^{(t)}\right)_{i=1}^{|V|} := \text{GNN}_\theta\left(\left(v_i^{(t)}\right)_{i=1}^{|V|}, E_t, t\right) \qquad \forall\, t = 0, \ldots, T-1,$

Temporal mixing: $\quad \left(\hat{v}_i^{(t)}\right)_{t=1}^{T} := \text{TransformerDecoder}_\theta\left(\left(\tilde{v}_i^{(t)}\right)_{t=1}^{T}\right) \qquad \forall\, i = 1, \ldots, |V|.$

For the structural mixing, we use a Structure-Aware-Transformer layer (Chen et al., 2022). Additionally, we incorporate both the timestep $t$ and cycle counts in $G_t$ using FiLM (Perez et al., 2018). These structural features were used previously in other graph generative models such as DiGress (Vignac et al., 2023). Multiple mixing operations are stacked to form the backbone model.

**Edge Decoder.** To model $p_\theta(G_t | G_{t-1}, \ldots, G_0)$, we produce a distribution over possible edge sets of $G_t$. We use a mixture of multivariate Bernoulli distributions to capture dependencies between

edges, similar to previous works (Liao et al., 2019; Kong et al., 2023). Given $K \geq 1$ mixture components, we infer $K$ Bernoulli parameters for each node pair $i, j \in V$ from the node representations $v_i$ produced by the backbone model:

$$p_k^{(i,j)} := D_{k,\theta}(v_i, v_j) \in [0, 1], \qquad \forall\, i, j \in V, \quad \forall\, 1 \leq k \leq K. \tag{5}$$

where $D_{\cdot,\theta}$ is some neural network. We enforce that $p_k^{(i,j)}$ is symmetric and that the probability of self-loops is zero. In addition, we produce a mixture distribution $\pi \in \mathbb{R}^K$ in the $K-1$ dimensional probability simplex from pooled node representations: $\pi := D_{\text{mix},\theta}\left((v_i)_{i=1}^{|V|}\right) \in \Delta^{K-1}$. The architectural details of $D_{\cdot,\theta}$ are provided in Appendix A.7. The final likelihood is defined as:

$$p_\theta(E_t | G_{t-1}, \dots, G_0) := \sum_{k=1}^{K} \pi_k \prod_{i<j} \begin{Bmatrix} p_k^{(i,j)} & \text{if} & e_{ij} \in E_t \\ 1 - p_k^{(i,j)} & \text{else} \end{Bmatrix}. \tag{6}$$

In contrast to existing autoregressive graph generators (You et al., 2018b; Liao et al., 2019; Goyal et al., 2020; Bacciu et al., 2020; Kong et al., 2023), *our model introduces a key innovation: the ability to generate non-monotonic graph sequences.* This means it can both add and delete edges. We argue that this capability is crucial for mitigating error accumulation during sampling. Consider, for instance, the task of generating tree structures. If a cycle is inadvertently introduced into an intermediate graph $G_t$ (where $t < T$), traditional autoregressive approaches would be unable to rectify this error. Our model, however, can potentially delete the appropriate edges in subsequent timesteps, thus recovering from such mistakes. In Sec. 3.3.1, we introduce a data augmentation technique to train AFM on non-monotonic sequences that contain erroneous edges and we show empirically in Sec. 4.4 that this augmentation substantially improves model performance.

**Input Node Representations.** The initialization of node representations is a crucial step preceding the forward pass through the mixer architecture above. We compute initial node representations from positional and structural features in a similar fashion as Vignac et al. (2023). Moreover, we add learned positional embeddings based on a node ordering derived from the filtration function. We refer to Appendix A.6 for further details.

**Asymptotic Complexity.** We provide a detailed analysis of the asymptotic runtime complexity of our method in Appendix A.2. Asymptotically, AFM's complexity of sampling a graph with $N$ nodes is $\mathcal{O}(T^2 N + T N^3)$. Even though cubic in the number of nodes, we found that the efficiency of AFM is largely driven by our ability to use a small $T$ ($T \leq 30$), while diffusion-based models generally require a much larger number of iterations.

## 3.3 TRAINING ALGORITHM

We employ the teacher-forcing approach (Williams & Zipser, 1989) to train our generative model $p_\theta$ in a first training stage. We illustrate this training scheme in Figure 1b. Teacher-forcing allows the model to learn from complete sequences of graph evolution, providing a good initialization for subsequent reinforcement learning-based fine-tuning (second training stage). Given a dataset of graphs $\mathcal{D}$, we convert it into a dataset of filtration sequences, denoted as $\tilde{\mathcal{D}}$. Our objective is to maximize the log-likelihood of these sequences under our model:

$$\mathcal{L}(\theta) := \mathbb{E}_{(G_0, \dots, G_T) \sim \tilde{\mathcal{D}}} \left[ \log p_\theta(G_0, \dots, G_T) \right]. \tag{7}$$

In practice, this objective is implemented as a cross-entropy loss. While the teacher-forcing approach is efficient, it can lead to exposure bias (Bengio et al., 2015; Yu et al., 2017; Ranzato et al., 2016), where the model's performance during inference degrades due to a distribution shift caused by its reliance on its own predictions. We propose two strategies to address this issue, namely noise augmentation and adversarial fine-tuning with reinforcement learning.

### 3.3.1 MITIGATING EXPOSURE BIAS

**Noise Augmentation.** To mitigate exposure bias in autoregressive modeling, previous works have proposed data augmentation schemes to make models more robust to the distribution-shift occuring

during inference (Bengio et al., 2015). We propose a simpler yet effective strategy: namely, randomly perturbing intermediate graphs in a filtration sequence $G_0, \ldots, G_T$ during the above teacher-forcing training phase to expose the model to erroneous transitions. For each intermediate graph $G_t$ with $0 < t < T$, we generate a perturbed edge set $\tilde{E}_t$ by including each possible edge $e$ independently with probability

$$\mathbb{P}[e \in \tilde{E}_t] := \begin{cases} (1 - \lambda_t) + \lambda_t \rho_t & \text{if} \quad e \in E_t \\ \lambda_t \rho_t & \text{else} \end{cases}, \tag{8}$$

where $\lambda_t \in [0, 1]$ controls stochasticity and $\rho_t := |E_t|/\binom{|V|}{2}$ is the density of $G_t$. In practice, we decrease $\lambda_t$ affinely as $t$ increases and include multiple perturbations of each filtration sequence in the training dataset $\tilde{\mathcal{D}}$. Regarding the choice of these hyper-parameters, we refer to Appendix A.8.

**Adversarial Fine-tuning with Reinforcement Learning.** While the above noise augmentation technique substantially improves the overall quality of generated graphs, it still falls short in generating graphs with high structural fidelity. To address this, we propose a reinforcement learning based fine-tuning stage to refine the model trained with teacher-forcing. Adapting the SeqGAN framework (Yu et al., 2017), we implement a generator-discriminator architecture where the generator (our mixer model) operates in inference mode as a stochastic policy and is thereby exposed to its own predictions during training. The discriminator is a graph transformer, namely GraphGPS (Rampásek et al., 2022). During training, the generator produces graph samples, which the discriminator evaluates for plausibility. The generator is updated using Proximal Policy Optimization (PPO) (Schulman et al., 2017) based on the discriminator's feedback, while the discriminator is trained adversarially to distinguish between generated and training set graphs. This training scheme is illustrated in Figure 1c. It is worth noting that only the final generated graph $G_T$ is presented to the discriminator, instead of the full sequence of graphs. Therefore, the generator is trained to maximize a terminal reward without constraints on intermediate graphs. We provide the pseudo-code in Appendix A.18.

## 4 EXPERIMENTS

We empirically evaluate our method on synthetic and real-world datasets. In Sec. 4.1, we first present results on the commonly used small benchmark datasets (Martinkus et al., 2022), comparing our method to a variety of baselines. We then demonstrate in Sec. 4.2 that we can improve upon these results by using a more realistic setting with more training examples. Additionally, we present results for inference efficiency. Finally, in Sec. 4.3, we demonstrate that our model is applicable to real-world data, namely larger protein graphs (Dobson & Doig, 2003). In Sec. 4.4, we present ablation studies demonstrating the efficacy of noise augmentation and adversarial fine-tuning.

**Evaluation.** We follow established practices from previous works (You et al., 2018b; Martinkus et al., 2022; Vignac et al., 2023) in our evaluation. We compare a set of model-generated samples to a test set via maximum mean discrepancy (MMD) (Gretton et al., 2012), based on various graph descriptors. These descriptors include histograms of node degrees (Deg.), clustering coefficients (Clus.), orbit count statistics (Orbit), and eigenvalues (Spec.). While we employ these metrics to facilitate comparison with previous methods, we acknowledge the criticisms raised by O'Bray et al. (2022) regarding the use of indefinite kernels and arbitrary selection of kernel hyperparameters in these evaluation techniques.

In previous works (Martinkus et al., 2022; Vignac et al., 2023), very few samples are generated for the evaluation of graph generative models. In Appendix A.17, we show both theoretically and empirically that this leads to high bias and variance in the reported metrics. In Sec. 4.2 and 4.3, we generate 1024 samples for evaluation to mitigate this issue, while we generate 40 samples in Sec. 4.1 to fairly compare to previous methods. We sample the number of nodes to be generated from the empirical training distribution, which is consistent with Vignac et al. (2023) but deviates from the approach by Bergmeister et al. (2024), where the authors determine the number of nodes by using the ground truth number of nodes from the test set. For synthetic datasets, we follow previous works by reporting the ratio of generated samples that are valid, unique, and novel (VUN). In Sec. 4.2 and 4.3, we report inference speed, measured as the time needed to generate 1024 graphs on an H100 GPU, normalized to a per-graph cost.

Table 1: Performance of various models on small synthetic SPECTRE datasets. Results on GraphRNN, GRAN and SPECTRE taken from Martinkus et al. (2022). Results on DiGress, ESGG and EDGE from Bergmeister et al. (2024).

| | Planar Graphs ($|V| = 64$, $N_{\text{train}} = 128$) | | | | | | SBM Graphs ($|V| \sim 104$, $N_{\text{train}} = 128$) | | | | |
|---|---|---|---|---|---|---|---|---|---|---|---|
| | VUN (↑) | Deg. (↓) | Clus. (↓) | Orbit (↓) | Spec. (↓) | | VUN (↑) | Deg. (↓) | Clus. (↓) | Orbit (↓) | Spec. (↓) |
| GraphRNN | 0.0 | 0.0049 | 0.2779 | 1.2543 | 0.0459 | GraphRNN | 5.0 | 0.0055 | 0.0584 | 0.0785 | 0.0065 |
| GRAN | 0.0 | 0.0007 | 0.0426 | 0.0009 | 0.0075 | GRAN | 25.0 | 0.0113 | 0.0553 | 0.0540 | 0.0054 |
| SPECTRE | 25.0 | 0.0005 | 0.0785 | 0.0012 | 0.0112 | SPECTRE | 52.5 | 0.0015 | 0.0521 | 0.0412 | 0.0056 |
| DiGress | 77.5 | 0.0007 | 0.0780 | 0.0079 | 0.0098 | DiGress | 60.0 | 0.0018 | 0.0485 | 0.0415 | 0.0045 |
| EDGE | 0.0 | 0.0761 | 0.3229 | 0.7737 | 0.0957 | EDGE | 0.0 | 0.0279 | 0.1113 | 0.0854 | 0.0251 |
| ESGG | 95.0 | 0.0005 | 0.0626 | 0.0017 | 0.0075 | HiGen | N/A | 0.0019 | 0.0498 | 0.0352 | 0.0046 |
| | | | | | | ESGG | 45.0 | 0.0119 | 0.0517 | 0.0669 | 0.0067 |
| Ours | 72.5 | 0.0037 | 0.1332 | 0.0047 | 0.0099 | Ours | 47.5 | 0.0014 | 0.0506 | 0.0551 | 0.0058 |

**Baselines.** We aim to demonstrate that our method is competitive with state-of-the-art diffusion models in terms of sample quality while outperforming them in terms of inference speed. Hence, we compare our method to two recent diffusion models, namely DiGress (Vignac et al., 2023) and ESGG (Bergmeister et al., 2024). DiGress first introduced discrete diffusion to the area of graph generation and remains one of the most robust baselines. ESGG is acutely relevant to our work, as it aims to improve inference speed and scalability to large graphs. In addition to these diffusion-based approaches, we also present results on an autoregressive model, GRAN (Liao et al., 2019), which focuses on efficiency during inference. Whenever we train baseline models, we continue training until no additional improvements in validation validitiy and MMD metrics are apparent. We provide additional details about model selection for ESGG in Appendix A.11 and for GRAN in Appendix A.15. We provide our hyper-parameter choices for GRAN in Appendix A.12, for DiGress in Appendix A.13, and for ESGG in Appendix A.14. In Sec. 4.1, we report baseline results from the literature, also comparing to the hierarchical HiGen (Karami, 2024) approach, the scalable EDGE (Chen et al., 2023) diffusion model, the autoregressive GraphRNN model (You et al., 2018b), and the GAN-based SPECTRE model (Martinkus et al., 2022).

## 4.1 EXPERIMENTS WITH SMALL SYNTHETIC DATASETS

As a first demonstration of our method, we present results on the planar and SBM datasets by Martinkus et al. (2022). Since the training set consists of only 128 graphs, we find that our models tend to overfit during the teacher-forcing training stage, which manifests as an increase in the validation loss while the evaluation metrics continue to improve. To mitigate this issue, we introduce some small stochastic perturbations to node orderings used for initializing node representations. We discuss this in more detail in Appendix A.6. Model selection is performed based on the minimal validation loss. Table 1 illustrates that our model outperforms GraphRNN (You et al., 2018b), GRAN (Liao et al., 2019), and SPECTRE (Martinkus et al., 2022), in terms of validity on the planar graph dataset. On the SBM dataset, it outperforms the autoregressive baselines (GraphRNN and GRAN) while almost matching the performance of SPECTRE. On both datasets, it is competitive with the two diffusion-based approaches, DiGress (Vignac et al., 2023) and ESGG (Bergmeister et al., 2024).

## 4.2 EXPERIMENTS WITH EXPANDED SYNTHETIC DATASETS

We supplement the results presented above by training our model on larger synthetic datasets. Namely, we generate training sets consisting of 8192 graphs and corresponding validation and test sets consisting of 256 graphs each. We use the same data generation approach as in Martinkus et al. (2022) to obtain expanded planar and SBM datasets. Additionally, we produce an expanded dataset of lobster graphs using NetworkX (Hagberg et al., 2008), as done in (Liao et al., 2019). To assess the robustness of our method, we perform three independent training runs per dataset. We present the median metrics along with the maximum deviations observed across the three runs in Appendix A.9 and visualize samples from our model in Appendix A.10. In Table 2, we compare our method to our three baselines. For reasons of brevity, we only report the median performance of our method here. We find that our models are substantially faster during inference than the diffusion models, consistently achieving at least a 100-fold speedup in comparison to DiGress and ESGG. In an independent experiment, we observe that reducing the number of diffusion steps in DiGress to values comparable to the ones used in AFM (30 or 15) leads to a substantial degradation of quality. Moreover, in Ta-

Table 2: Performance of various models on expanded synthetic datasets, evaluated on 1024 model samples. We report the result across a single run for the baselines and the median performance across three runs for our model. *ESGG evaluation is modified to draw graph sizes from empirical training distribution and use 100 refinement steps for determining validity.

| | Expanded Planar Graphs ($|V| = 64$, $N_{\text{train}} = 8192$) | | | | | | | |
|---|---|---|---|---|---|---|---|---|
| | VUN ($\uparrow$) | Unique ($\uparrow$) | Novel ($\uparrow$) | Deg. ($\downarrow$) | Clus. ($\downarrow$) | Orbit ($\downarrow$) | Spec. ($\downarrow$) | Time ($\downarrow$) |
| GRAN | 0.19 | 100 | 100 | 0.0061 | 0.1862 | 0.0961 | 0.0081 | 0.0303 |
| DiGress | 80.76 | 100 | 100 | 0.0004 | 0.0217 | 0.0045 | 0.0024 | 2.73 |
| ESGG* | 89.94 | 100 | 100 | 0.0007 | 0.0162 | 0.0074 | 0.0012 | 4.65 |
| Ours | 79.20 | 100 | 100 | 0.0004 | 0.0183 | 0.0002 | 0.0012 | 0.0278 |
| | Expanded SBM Graphs ($N_{\text{train}} = 8192$) | | | | | | | |
| | VUN ($\uparrow$) | Unique ($\uparrow$) | Novel ($\uparrow$) | Deg. ($\downarrow$) | Clus. ($\downarrow$) | Orbit ($\downarrow$) | Spec. ($\downarrow$) | Time ($\downarrow$) |
| GRAN | 25.29 | 100 | 100 | 0.0186 | 0.0086 | 0.0305 | 0.0022 | 0.133 |
| DiGress | 56.15 | 100 | 100 | 0.0002 | 0.0056 | 0.0076 | 0.0009 | 12.99 |
| ESGG* | 3.52 | 100 | 100 | 0.0949 | 0.0121 | 0.0518 | 0.0122 | 39.42 |
| Ours | 75.98 | 100 | 100 | 0.0014 | 0.0051 | 0.0180 | 0.0011 | 0.0301 |
| | Expanded Lobster Graphs ($N_{\text{train}} = 8192$) | | | | | | | |
| | VUN ($\uparrow$) | Uniqe ($\uparrow$) | Novel ($\uparrow$) | Deg. ($\downarrow$) | Clus. ($\downarrow$) | Orbit ($\downarrow$) | Spec. ($\downarrow$) | Time ($\downarrow$) |
| GRAN | 41.99 | 99.90 | 97.56 | 0.0436 | 0.0069 | 0.1510 | 0.1469 | 0.0399 |
| DiGress | 96.58 | 99.22 | 96.78 | 0.0001 | $8.33 \times 10^{-7}$ | 0.0016 | 0.0009 | 4.86 |
| ESGG* | 63.96 | 99.61 | 98.24 | 0.0007 | 0.00 | 0.0027 | 0.0023 | 3.16 |
| Ours | 79.10 | 99.80 | 100 | 0.0004 | $7.89 \times 10^{-5}$ | 0.0010 | 0.0016 | 0.0297 |

Table 3: Performance of various models on protein graph dataset. *ESGG evaluation is modified to draw graph sizes from empirical training distribution.

| | Protein Graphs ($100 \leq |V| \leq 500$, $N_{\text{train}} = 587$) | | | | |
|---|---|---|---|---|---|
| | Deg. ($\downarrow$) | Clus. ($\downarrow$) | Orbit ($\downarrow$) | Spec. ($\downarrow$) | Time ($\downarrow$) |
| GRAN | 0.0025 | 0.0510 | 0.1539 | 0.0051 | 2.25 |
| DiGress | 0.0006 | 0.0234 | 0.0289 | 0.0014 | 72.27 |
| ESGG* | 0.0033 | 0.0216 | 0.0557 | 0.0008 | 19.48 |
| Ours | 0.0024 | 0.0464 | 0.0532 | 0.0024 | 0.194 |

ble 2 we find that our method appears competitive with respect to sample quality, outperforming the two diffusion models on the expanded SBM dataset in terms of validity. For ESGG, we note that we obtain a surprisingly low validity score on the expanded SBM dataset. We refer to Appendix A.11 for further discussion on this. In comparison to the autoregressive baseline, GRAN, we find that our model substantially outperforms it in terms of validity and MMD metrics.

## 4.3 EXPERIMENTS WITH REAL-WORLD DATA

In this subsection, we present empirical results on the protein graph dataset introduced by Dobson & Doig (2003). While results have been reported for this dataset in previous works, we re-evaluate the baselines on 1024 model samples to reduce bias and variance in the reported metrics. We use a trained GRAN checkpoint provided by Liao et al. (2019) but re-train ESGG and DiGress, as no trained models are available. We find that our model is again substantially faster than the diffusion-based baselines. Moreover, it is also 10 times faster than GRAN while outperforming it with respect to all MMD metrics. In comparison to the diffusion-based models, the sample quality of our approach appears slightly worse by most MMD metrics.

## 4.4 ABLATION STUDIES

In this subsection, we present empirical results which demonstrate that the noise augmentation of intermediate graphs and adversarial fine-tuning introduced in Sec. 3.3 are crucial components of our approach. We understand this as a strong indication that exposure bias affects our autoregressive model. Additionally, we study the impact of the filtration granularity, as determined by the hyper-

Table 4: Two ablation studies on expanded planar graph dataset. Results with median $\pm$ maximum deviation across three runs are reported. For the noise ablation, we train for 100k steps in stage I. For the finetuning ablation, we train for 200k steps in stage I.

| | Noise Ablation | | Finetuning Ablation | |
|---|---|---|---|---|
| | Stage I w/ Noise | Stage I w/o Noise | Stage II | Stage I w/ Noise |
| VUN ($\uparrow$) | **20.21** $\pm$ 3.22 | 0.00 $\pm$ 0.00 | **79.20** $\pm$ 7.13 | 23.24 $\pm$ 9.67 |
| Degree ($\downarrow$) | **0.0058** $\pm$ 0.0008 | 0.0864 $\pm$ 0.0749 | **0.0004** $\pm$ $5.4256 \times 10^{-5}$ | 0.0036 $\pm$ 0.0009 |
| Clustering ($\downarrow$) | **0.1768** $\pm$ 0.0106 | 0.3179 $\pm$ 0.0037 | **0.0183** $\pm$ 0.0014 | 0.1547 $\pm$ 0.0280 |
| Spectral ($\downarrow$) | **0.0048** $\pm$ 0.0011 | 0.1042 $\pm$ 0.0760 | **0.0012** $\pm$ 0.0004 | 0.0033 $\pm$ 0.0014 |
| Orbit ($\downarrow$) | **0.0129** $\pm$ 0.0169 | 0.7115 $\pm$ 0.4411 | **0.0002** $\pm$ 0.0016 | 0.0043 $\pm$ 0.0023 |

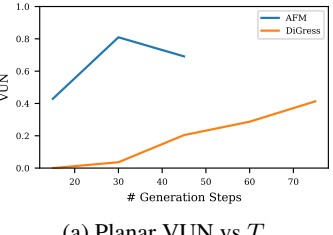

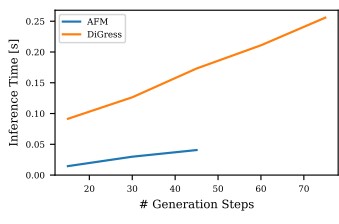

(a) Planar VUN vs $T$        (b) Inference time vs $T$

Figure 2: Performance and inference speed of AFM and DiGress on the expanded planar graph dataset as the number of generation steps is varied.

parameter $T$. In Appendix A.16, we present extensive additional ablations on the choice of filtration function, filtration schedule, and node individualization.

**Noise Augmentation.** Empirically, we find that noise augmentation of intermediate graphs substantially improves performance during training with teacher forcing (stage I). We illustrate this on the expanded planar graph dataset in the left half of Table 4. As we consider this an important finding of our work, we perform three training runs with different seeds for this ablation.

**GAN Tuning.** In the right half of Table 4, we compare performance after training with teacher-forcing (stage I) and adversarial fine-tuning (stage II) on the expanded planar graph dataset from Sec. 4.2. We find that adversarial fine-tuning substantially improves performance, both in terms of validity and MMD metrics. A corresponding analysis for the expanded SBM and lobster datasets can be found in Appendix A.16.

**Filtration Granularity.** In Figure 2, we study the impact of filtration granularity, *i.e.,* the number of steps $T$, on the generation quality and inference speed of AFM for the expanded planar graph dataset. Additionally, we investigate how the number of denoising steps influences the quality and speed in DiGress. We systematically re-train the models with varying $T$. Notably, AFM consistently outperforms DiGress in computational efficiency across all steps. While DiGress only achieves a maximum VUN of 41% for our largest considered $T$, AFM achieves a VUN of 81% for $T = 30$.

## 5 CONCLUSION

We proposed AFM, an efficient autoregressive graph generative model that relies on graph filtration. AFM generates high-quality graphs, outperforming existing autoregressive models and rivaling discrete diffusion approaches in terms of quality while being substantially faster at inference. Various ablations demonstrated the configurability of AFM and indicate that exposure bias is an important challenge for autoregressive graph modeling.

One limitation lies in the focus of our methodology solely on generating non-attributed graphs. While extending this approach to include categorical edge labels seems feasible (see Appendix A.7), the incorporation of node labels presents a bigger challenge. Although post-processing techniques for node labeling may be applicable, the direct modeling of attributes remains a crucial area for future investigation. Furthermore, exploring the possibility of learning to reverse a node label filtration process jointly with the edge filtration could be a promising direction for future research.

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

# A APPENDIX

## A.1 EXTENDED RELATED WORK

In this section, we extend Sec. 2 and provide additional comparative analyses to previous works.

**Other Graph Generative Models.** In concurrent work, Zhao et al. (2024) introduce a hybrid graph generative model, termed Pard, combining autoregressive and discrete diffusion components. Similar to AFM, Pard generates graphs by building a sequence of increasingly large sub-graphs. In contrast to the method we present here, Pard is limited to the generation of *induced* sub-graphs and uses a shared diffusion model to sample them. While the authors state efficiency as one motivation for their approach, they do not present runtime measurements during inference.

**Graph Diffusion.** Similar to graph diffusion models (Vignac et al., 2023), we propose a corrupting process to transform graph samples $G_T$ into graphs $G_0$ from some convergent distribution (in our case the point-mass at the empty graph). However, in contrast to denoising diffusion models, the process we are proposing is not Markov.

**Absorbing State Diffusion.** Absorbing state graph diffusion (Chen et al., 2023; Kong et al., 2023) resembles our approach in that it also generates a sequence of increasingly dense graphs. We aim to increase efficiency by generating substantially shorter sequences than previous works. In practice, we choose to generate graphs within 15 or 30 steps. EDGE (Chen et al., 2023) requires between 64 and 512 denoising steps, depending on the dataset. To generate a graph on $N$ nodes, GraphARM (Kong et al., 2023) requires $N$ denoising steps, as exactly one node decays to the absorbing state at a time. The larger number of denoising steps in these methods may increase inference time and necessitates a first-order autoregressive structure. Additionally, as detailed above, our proposed method does not readily fit into the framework of denoising diffusion models.

## A.2 COMPLEXITY ANALYSIS

In the following, we analyze the asymptotic runtime complexity of sampling a graph from our proposed model and the baselines we studied in Section 4.

**Proposition 1.** *The asymptotic runtime complexity for sampling a graph with $N$ nodes from an AFM with $T$ timesteps is:*

$$\mathcal{O}(T^2 N + T N^3) \tag{9}$$

*Proof.* To sample a graph from an AFM, one has to perform $T$ forward passes through our proposed mixer architecture. These forward passes are preceded by the computation of various graph features, including laplacian eigenvalues and eigenvectors. This eigendecomposition has complexity $\mathcal{O}(N^3)$. At timestep $0 \leq t < N$, the structural mixing layers have complexity $\mathcal{O}(N^2)$ due to the self-attention component of SAT. The temporal mixing layers, on the other hand, have complexity $\mathcal{O}(N(t+1))$, as each node attends to its representations at timesteps $0, \ldots, t$. We bound this complexity by $\mathcal{O}(NT)$. Hence, aggregating these complexities across all $T$ timesteps, we obtain the following runtime complexity:

$$\mathcal{O}(T^2 N + T N^2 + T N^3) = \mathcal{O}(T^2 N + T N^3) \tag{10}$$

$\square$

Below we show that the asymptotic complexity of AFM differs from the complexity of DiGress only in the quadratic term w.r.t. $T$:

**Proposition 2.** *The asymptotic runtime complexity for sampling a graph with $N$ nodes from a DiGress model with $T$ denoising steps is:*

$$\Omega(T N^3) \tag{11}$$

*Proof.* Similar to AFM, DiGress performs an eigendecomposition of the graph laplacian in each denoising step. Hence, one obtains a complexity of $\Omega(N^3)$ in each timestep, resulting in an overall complexity of $\Omega(T N^3)$. $\square$

We further analyze the asymptotic complexities of our other baselines of Sec. 4.2 and Sec. 4.3.

**Proposition 3.** *The asymptotic runtime complexity for sampling a graph with $N$ nodes from a GRAN model is $\Omega(N^2)$.*

*Proof.* GRAN explicitly constructs a dense adjacency matrix with $N^2$ entries. $\square$

**Proposition 4.** *The asymptotic runtime complexity of sampling a graph with $N$ nodes and $M$ edges from an ESGG model is $\Omega(N + M)$.*

*Proof.* This bound should trivially be satisfied by any generative model, as one already needs $\Omega(N + M)$ bits to represent a graph with $M$ edges on $N$ nodes. We refer to (Bergmeister et al., 2024) for a discussion on how tight this bound is. $\square$

In Table 5, we summarize these asymptotic complexities. While this analysis may suggest that AFM

Table 5: Asymptotic runtime complexities for sampling from different graph generative models.

| Method | Sampling complexity |
|---|---|
| AFM | $\mathcal{O}(T^2 N + T N^3)$ |
| DiGress | $\Omega(T N^3)$ |
| GRAN | $\Omega(N^2)$ |
| ESGG | $\Omega(N + M)$ |

does not scale well to extremely large graphs, we caution the reader that the asymptotic behavior may not accurately reflect efficiency in practice: Firstly, multiplicative constants and lower-order terms are ignored. Hence, it remains unknown in which regimes the asymptotic behavior governs inference time. Secondly, the analysis was made under the assumption that hyper-parameter choices (i.e. depth, width, etc.) is kept constant as $N$ and $M$ increase. It is reasonable to expect that more expressive networks are required to model large graphs.

### A.3 A First-Order Autoregressive Variant

As we demonstrated in Appendix A.2, the runtime of AFM is quadratic in the number of generation steps $T$ due to the temporal mixing operations which are implemented as transformer decoder layers. Analogously, one may verify that the memory complexity of sampling from AFM is linear in $T$. In this subsection, we study a simplified variant of AFM in which we use a first-order autoregressive structure. I.e., we enforce:

$$p_\theta(G_{t+1}|G_t, \ldots, G_0) = p_\theta(G_{t+1}|G_t) \tag{12}$$

We implement this by ablating the causally masked self-attention mechanism from the transformer layers in our mixer model, leaving only the feed-forward modules. The resulting first-order variant of AFM has space complexity which is independent of $T$ and runtime complexity which is linear in $T$.

We train such a first-order variant of AFM on the expanded planar graph dataset, using the same hyperparameters as for the transformer-based variant (see Appendix A.8). Using the first-order variant, we observe training instabilities after the first 100k training steps of stage I. While reducing the learning rate rectifies this instability, we find that this slows learning progress substantially. Instead, we use a model checkpoint at 100k steps and continue with training stage II.

In Table 6, we compare the performance of the transformer-based and the first-order variants after 100k steps of stage I training. In Table 7, we compare the performance after the subsequent stage II training. While we perform only 100k training steps in stage I for the first-order variant, we perform 200k training steps for the transformer-based variant, as it did not exhibit instabilities.

Generally, we observe that the transformer-based AFM variant slightly outperforms the first-order variant in terms of quality. However, the first-order variant remains competitive after stage II trainig and, thus, may be a suitable alternative in cases where a large $T$ is chosen. In our setting ($T = 30$), however, we find that the first-order variant is not substantially faster during inference, indicating that the runtime is not governed by the quadratic complexity in $T$.

Table 6: Performance of two AFM variants on the expanded planar graph dataset after 100k steps of stage I training. Showing median across three runs for the transformer-based variant and a single run for the first-order variant. All models reach perfect uniqueness and novelty scores.

| | VUN ($\uparrow$) | Deg. ($\downarrow$) | Clus. ($\downarrow$) | Orbit ($\downarrow$) | Spec. ($\downarrow$) |
|---|---|---|---|---|---|
| Transformer | 20.21 | 0.0058 | 0.1768 | 0.0129 | 0.0048 |
| First-Order | 5.66 | 0.0004 | 0.1782 | 0.0041 | 0.0035 |

Table 7: Performance of two AFM variants on the expanded planar graph dataset after stage II training. The transformer-based variant was trained for 200k steps in stage I while the first-order variant was trained for 100k steps in stage I. Showing median across three runs for the transformer-based variant and a single run for the first-order variant. All models reach perfect uniqueness and novelty scores.

| | VUN ($\uparrow$) | Deg. ($\downarrow$) | Clus. ($\downarrow$) | Orbit ($\downarrow$) | Spec. ($\downarrow$) | Time ($\downarrow$) |
|---|---|---|---|---|---|---|
| Transformer | 79.20 | 0.0004 | 0.0183 | 0.0002 | 0.0012 | 0.0278 |
| First-Order | 70.02 | 0.0004 | 0.0229 | 0.0046 | 0.0013 | 0.0247 |

### A.4 A Bound on Model Evidence

Given a graph $G_T$, let

$$q(G_{T-1}, \ldots G_1 | G_T) = \prod_{t=1}^{T-1} q(G_t | G_T) \tag{13}$$

be the data distribution over filtrations of this graph, determined by the choice of filtration function, scheduling, and noise augmentation. We assume that $G_0$ is deterministically the completely disconnected graph. Moreover, we note that by applying our noise augmentation strategy we ensure that $q$ is supported everywhere. Given some graph $G_T$, we can now derive the following evidence lower bound:

$$
\begin{aligned}
\log p_\theta(G_T) &= \sum_{G_1, \ldots, G_{T-1} \in \mathcal{G}} p_\theta(G_T, \ldots, G_0) \\
&= \log \sum_{G_1, \ldots, G_{T-1} \in \mathcal{G}} q(G_{T-1}, \ldots, G_1 | G_T) \frac{p_\theta(G_T, \ldots, G_0)}{q(G_{T-1}, \ldots, G_1 | G_T)} \\
&\geq \mathbb{E}_{q(\cdot | G_T)} \left[ \log \frac{p_\theta(G_T, \ldots, G_0)}{q(G_{T-1}, \ldots, G_1 | G_T)} \right] \\
&= \mathbb{E}_q \left[ \log p_\theta(G_T | G_{T-1}, \ldots, G_0) + \sum_{t=1}^{T-1} \log p_\theta(G_t | G_{t-1}, \ldots, G_0) \right. \\
&\quad \left. - \log q(G_t | G_T) \right]
\end{aligned}
\tag{14}
$$

We note that this lower bound is (up to sign and a constant that does not depend on $\theta$) exactly the autoregressive loss we use in training stage I. Hence, while we train AFM to model sequences of graphs, we do actually optimize an evidence lower bound for the final graph samples $G_T$.

### A.5 Pracical Advice on Hyperparameter Choice

In the following, we provide some pracitcal advice on choosing some of the most important hyper-parameters in AFM. Generally, we tuned few hyper-parameters in our experiments. We found the number of generation steps $T$ to be one of the most impactful hyper-parameters.

**Filtration Function.** The filtration function $f : E \rightarrow \mathbb{R}$ is the main component determining the structure of the graph sequence during stage I training. As discussed in Sec. 3.1, we recommend

that $f$ should convey meaningful information about the structure (i.e., be structurally consistent) and assign (mostly) distinct values to distinct edges (i.e., be diverse). We note that if $f$ fails to be diverse, many edges may be added in a single generation step, regardless of the choice of $T$. We found the edge Fiedler function to perform well in many settings, and used this filtration function throughout our experiments. We recommend that practioners utilize this filtration function and perform experiments with further filtration functions that incorporate domain-specific inductive biases. In the case of generating protein graphs, for instance, one may consider a filtration function that quantifies the distance of residues in the sequence (this filtration would first generate a backbone path, followed by increasingly long-range interactions of residues).

**Filtration Granularity.** As we demonstrate in Sec. 4.4, the choice of the number of generation steps $T$ has a substantial influence on sampling efficiency and generation. Generally, $T$ can be chosen substantially smaller than in other autoregressive models. In our experiments, we chose $T = 15$ or $T = 30$. We recommend that practitioners experiment with different values in this order of magnitude. We further caution that increasing $T$ does not necessarily improve sample quality, and may actually harm it.

**Scheduling Function.** The scheduling function $\gamma : [0, 1] \to [0, 1]$ governs the rate at which edges are added at different timesteps and should be monotonically increasing with $\gamma(0) = 0$ and $\gamma(1) = 1$. We found the heuristic choice of $\gamma(t) := t$ to work well in many settings. However, as we demonstrate in Appendix A.16, the concave schedule may be a promising alternative. We recommend that practitioners validate stage I training with a convex, linear, and concave scheduling function. We note that the scheduling function is no longer used during stage II training, as the model is left free to generate arbitrary intermediate graphs. Hence, performance after stage I training may be a suitable metric for selecting a scheduling function.

**Noise Augmentation.** We use noise augmentation during training stage I to counteract exposure bias, i.e. the accumulation of errors in the sampling trajectory. Manual inspection of the graph sequence $G_0, \ldots, G_T$ may be difficult. However, we found that inspecting the development of the edge density over this graph sequence can provide a simple tool for diagnosing exposure bias. Namely, we expect the density to be mostly governed by the scheduling function $\gamma(t)$. E.g., for the linear schedule, the density should increase roughly linearly with $t$. In models that do not utilize noise augmentation, we can observe that, after some generation steps, the edge density can oftemtimes deviate from this expected behavior (e.g. by suddenly increasing or becoming non-monotonic). In this case we expect that noise augmentation can rectify exposure bias. In our experiments, we find that we do not need to tune the noise schedule. Instead, we fix a single schedule that is shared across all models. For details on this schedule, we refer to Appendix A.8.

**Perturbation of Node Orderings.** During training stage I, AFM may overfit on small datasets. This manifests as an increase in validation loss, while the validation MMD metrics continue to improve. We observe this behavior only on the small datasets in Sec. 4.1 and find that it can be mostly attributed to the node ordering used to derive initial node representations (c.f. Appendix A.6). We recommend to monitor validation losses during stage I training. If the validation loss starts to slowly increase while the training loss continues to decrease, we recommend to randomly perturb the node ordering, as described in Appendix A.6. Increase the noise scale $\sigma$ until no over-fitting can be observed.

A.6 INPUT NODE REPRESENTATIONS

We define the input node representations as:

$$v_i^{(t)} := f_\theta(G_t)_i + W_i^{\text{node}}, \tag{15}$$

where $f_\theta$ produces node features from Laplacian positional encodings (Dwivedi et al., 2023), random walk positional encodings (Dwivedi et al., 2022), and cycle counts following DiGress (Vignac et al., 2023). The matrix $W^{\text{node}} \in \mathbb{R}^{N \times D}$ is a trainable embedding layer where $N$ denotes the cardinality of the largest vertex set seen during training. It is important to note that the computation of input node representations requires a specific node ordering. While the permutation equivariance of our model and the symmetry of the initially empty graph $G_0$ allow for arbitrary ordering during

inference, we employ a structured approach during teacher-forcing training. This ordering is derived from the structure of the final graph $G_T$ and is based on the filtration function $f$.

Specifically, we propose a node weighting scheme $h : V \to \mathbb{R}$ defined as:

$$h(i) := \frac{1}{|\mathcal{N}_G(i)|} \sum_{j \in \mathcal{N}_G(i)} f(e_{ij}), \qquad \forall\, i \in V, \tag{16}$$

where $\mathcal{N}_G(i)$ represents the neighborhood of node $i$ in $G$. This weighting assigns to each node the average weight of its incident edges, as determined by the filtration function $f$. We then establish a node ordering such that $h$ is non-increasing. The impact of different ordering strategies on model performance is further studied and compared in Appendix A.16.

When training on small datasets, such as those introduced by Martinkus et al. (2022), we find that the node individualization in Eqn. (15) can lead to overfitting. This manifests as an increase in validation loss, while the evaluation metrics (i.e. MMD and VUN) continue to improve. As a data augmentation strategy to avoid overfitting, we propose to add Gaussian noise to the node weights $h_G$ defined in Eqn. (15) when training on small datasets. I.e., we use the perturbed node weights

$$h_G(s) + \mathcal{N}(0, \sigma^2) \tag{17}$$

for sorting the nodes. We emphasize that this measure is independent of the perturbation of intermediate graphs introduced in Sec. 3.3. Moreover, we perturb node orderings only in the experiments on the small SPECTRE datasets (i.e., in Sec 4.1).

## A.7 EDGE DECODER ARCHITECTURE

In this subsection, we present details on the edge decoder $D_{\cdot, \theta}$. While our approach is in principle applicable to discretely labeled edges, we concentrate on predicting distributions over unlabeled edges here. Fix some timestep $0 \le t < T$. Assume that for this timestep, we are given some node representations $(v_i)_{i=1}^{|V|}$ produced by the backbone model. The edge decoder contains $K$ submodules that produce multivariate Bernoulli distributions. Assuming that the node-representations produced by the backbone are $D$-dimensional, let $\mathrm{Dense}_{k,\theta}^{(1)} : \mathbb{R}^D \to \mathbb{R}^{2D}$ and $\mathrm{Dense}_{k,\theta}^{(2)}, \mathrm{Dense}_{k,\theta}^{(3)} : \mathbb{R}^{2D} \to \mathbb{R}^{2D}$ be fully connected layers learned for each component $k$. Define corresponding MLPs:

$$\mathrm{MLP}_{k,\theta} := \mathrm{ReLU} \circ \mathrm{Dense}_{k,\theta}^{(2)} \circ \mathrm{ReLU} \circ \mathrm{Dense}_{k,\theta}^{(1)} \tag{18}$$

For each $k$, we process the node representations $v_i$ separately and split the resulting vectors into two $D$-dimensional halves:

$$(x_i^{(k)}, y_i^{(k)}) := \mathrm{MLP}_{k,\theta}(v_i) \qquad (\hat{x}_i^{(k)}, \hat{y}_i^{(k)}) := \mathrm{Dense}_k^{(3)}\left((x_i^{(k)}, y_i^{(k)})\right) \tag{19}$$

We define the logit $l_{k,i,j}$ for the presence of an edge and the logit $r_{k,i,j}$ for the absence of an edge:

$$l_{k,i,j} := \frac{x_i^{(k)\top} \hat{x}_j^{(k)} + x_j^{(k)\top} \hat{x}_i^{(k)}}{2} \qquad r_{k,i,j} := \frac{y_i^{(k)\top} \hat{y}_j^{(k)} + y_j^{(k)\top} \hat{y}_i^{(k)}}{2} \tag{20}$$

Finally, we define the likelihood of the presence of an edge as:

$$D_{k,\theta}(v_i, v_j) := \frac{\exp(l_{k,i,j})}{\exp(l_{k,i,j}) + \exp(r_{k,i,j})} \tag{21}$$

While this modeling of the mixture distributions is quite involved, it allows the edge decoder to be easily extended to produce distributions over labeled edges by producing logits for labels of node pairs (instead of producing logits for presence and absence of edges).

Finally, we compute a mixture distribution $\pi \in \Delta^{K-1}$ via $D_{\mathrm{mix},\theta}$. To this end, we learn a node-level MLP:

$$\mathrm{MLP}_{\mathrm{mix},\theta}^{(1)} := \mathrm{ReLU} \circ \mathrm{Dense}_{\mathrm{mix},\theta}^{(1)} \tag{22}$$

and a graph-level MLP:

$$\mathrm{MLP}_{\mathrm{mix},\theta}^{(2)} := \mathrm{Dense}_{\mathrm{mix},\theta}^{(3)} \circ \mathrm{ReLU} \circ \mathrm{Dense}_{\mathrm{mix},\theta}^{(2)} \tag{23}$$

where $\text{Dense}_{\text{mix},\theta}^{(3)} : \mathbb{R}^D \to \mathbb{R}^K$. We then define:

$$D_{\text{mix},\theta}\left((v_i)_{i=1}^{|V|}\right) := \text{softmax}\left(\text{MLP}_{\text{mix},\theta}^{(2)}\left(\frac{1}{|V|}\sum_{i=1}^{|V|}\text{MLP}_{\text{mix},\theta}^{(1)}(v_i)\right)\right) \tag{24}$$

In the following, we discuss how our approach, and the edge decoder in particular, may be extended to edge-attributed and directed graphs.

**Edge Attributes.** While we only present experiments on un-attributed graphs, we note that our approach (in particular the edge decoder) is naturally extendable to discretely edge-attributed graphs. Assuming that one has $S$ possible edge labels (where one edge label encodes the absence of an edge), one would predict $S$ logits $l_{k,i,j}^{(s)}$ instead of predicting only two logits $l_{k,i,j}$ and $r_{k,i,j}$. Then, for fixed $i, j, k$, the vector

$$\text{softmax}_s\left(l_{k,i,j}^{(s)}\right)_{s=1}^{S} \qquad \in \qquad \Delta^{S-1} \tag{25}$$

would provide a distribution over labels for edge $\{v_i, v_j\}$. This distribution would be incorporated into a mixture (over $k$) of categorical distributions as above. Eqn. (6) would be adjusted to quantify the likelihood of edge labels instead of the likelihood of edge presence/absence.

**Directed Graphs.** In our experiments, we only consider applications of AFM to undirected graphs. However, our approach is also naturally extendable to directed graphs. Concretely, one would first adjust all GNNs in AFM to take edge directionality into account. One would additionally modify the product in Eqn. (6) to run over the entire adjacency matrix instead of only considering the upper triangle. I.e., one would get:

$$p_\theta(E_t|G_{t-1},\dots,G_0) := \sum_{k=1}^{K}\pi_k\prod_{i,j}\begin{cases} p_k^{(i,j)} & \text{if} \quad e_{ij}\in E_t \\ 1-p_k^{(i,j)} & \text{else} \end{cases}. \tag{26}$$

Finally, the edge decoder would be adjusted in Eqn. (20) to drop the symmetrization of $l_{k,i,j}$ and $r_{k,i,j}$ w.r.t. $i$ and $j$ (i.e., one no longer enforces the presence of the edge $(v_i, v_j)$ to have the same probability as the presence of $(v_j, v_i)$).

## A.8 AFM Hyperparameters

In Table 8, we summarize the most important hyperparameters of the generative model used in our experiments, including the number of filtration steps ($T$), mixture components ($K$), learning rate (LR), batch size (BS) in the format `num_gpus × grad_accumulation × local_bs`, and the number of perturbed filtration sequences we produce per graph in our training set (# Perturbations). We use a linear schedule and the line Fiedler filtration function in all experiments, unless indicated otherwise.

Table 8: Hyper-parameters for generative model

| | SPECTRE Planar | SPECTRE SBM | Expanded Planar | Expanded SBM | Expanded Lobster | Protein |
|---|---|---|---|---|---|---|
| $T$ | 30 | 15 | 30 | 15 | 30 | 15 |
| $K$ | 8 | 4 | 8 | 4 | 8 | 16 |
| # Layers | | | 5 | | | |
| Hidden Dim | | | 256 | | | |
| Laplacian PE dim. | | | 4 | | | |
| RWPE dim. | | | 20 | | | |
| Noise Augm. | | $\lambda_t$ affine with $\lambda_1 = 0.25$ and $\lambda_{T-1} = 0.05$ | | | | |
| # Perturbations | 256 | 256 | 4 | 4 | 4 | 8 |
| Perturb Node Order | Yes | Yes | No | No | No | No |
| Stg. I LR | $2.5\times 10^{-5}$ | $1\times 10^{-5}$ | $2.5\times 10^{-5}$ | $1\times 10^{-5}$ | $1\times 10^{-5}$ | $1\times 10^{-5}$ |
| Stg. I BS | $2\times 1\times 32$ | $2\times 1\times 32$ | $2\times 1\times 32$ | $2\times 1\times 32$ | $2\times 1\times 32$ | $2\times 4\times 8$ |
| # Stg. I Steps | 50k | 100k | 200k | 200k | 100k | 100k |
| Stg. I Precision | | | BF16 AMP | | | |
| Stg. II LR | | | $1.25\times 10^{-7}$ | | | |
| Stg. II BS | $1\times 4\times 32$ | $1\times 4\times 32$ | $1\times 4\times 32$ | $1\times 4\times 32$ | $1\times 4\times 32$ | $1\times 16\times 8$ |
| # Stg. II Iters | 2.5k | 3k | 1.5k | 5k | 4k | 1.5k |

In Table 9, we additionally provide the most important hyperparameters of the discriminator and value model trained during the adversarial fine-tuning stage.

Table 9: Hyper-parameters of discriminator and value model used during adversarial fine-tuning.

| | | SPECTRE Planar | SPECTRE SBM | Expanded Planar | Expanded SBM | Expanded Lobster | Protein |
|---|---|---|---|---|---|---|---|
| Disc. | LR | | | $1.00 \times 10^{-4}$ | | | |
| | BS | | | $1 \times 1 \times 32$ | | | |
| | # Layers | 2 | 3 | 3 | 3 | 3 | 2 |
| | Hidden dim. | 32 | 128 | 128 | 128 | 128 | 64 |
| | RWPE dim. | 5 | 20 | 20 | 20 | 20 | 20 |
| Val. | LR | | | $2.50 \times 10^{-4}$ | | | |
| | BS | | | $1 \times 4 \times 32$ | | | |
| | # Layers | | | 5 | | | |
| | Hidden dim. | | | 128 | | | |

## A.9 COMPREHENSIVE EVALUATION RESULTS ON EXPANDED SYNTHETIC DATASETS

In Table 10, we present the deviations observed across the three training runs discussed in Sec. 4.2.

Table 10: Evaluation results for AFM trained on expanded synthetic datasets. Showing median across three runs $\pm$ maximum deviation.

| | Expaned Planar | Expanded SBM | Expanded Lobster |
|---|---|---|---|
| VUN ($\uparrow$) | $79.20 \pm 7.13$ | $75.98 \pm 3.71$ | $79.10 \pm 7.13$ |
| Degree ($\downarrow$) | $0.0004 \pm 5.43 \times 10^{-5}$ | $0.0014 \pm 0.0062$ | $0.0004 \pm 0.0013$ |
| Clustering ($\downarrow$) | $0.0183 \pm 0.0014$ | $0.0051 \pm 0.0009$ | $7.89 \times 10^{-5} \pm 5.32 \times 10^{-5}$ |
| Spectral ($\downarrow$) | $0.0012 \pm 0.0004$ | $0.0011 \pm 0.0006$ | $0.0016 \pm 0.0028$ |
| Orbit ($\downarrow$) | $0.0002 \pm 0.0016$ | $0.0180 \pm 0.0171$ | $0.0010 \pm 0.0156$ |
| Unique ($\uparrow$) | $100.00 \pm 0.00$ | $100.00 \pm 0.00$ | $99.80 \pm 0.10$ |
| Novel ($\uparrow$) | $100.00 \pm 0.00$ | $100.00 \pm 0.00$ | $100.00 \pm 0.10$ |

## A.10 QUALITATIVE MODEL SAMPLES

In Figure 3, we present uncurated samples from the different models described in Sec. 4.

## A.11 ESGG MODEL SELECTION

While ESGG maintains exponential moving averages of model weights during training, we choose to only evaluate non-smoothed model weights (i.e. the EMA weights with decay parameter $\gamma = 1$), as validation is compute-intensive.

**SBM Dataset.** In our experiments, we obtain worse performance on the expanded SBM dataset than was reported on the smaller SPECTRE SBM dataset in (Bergmeister et al., 2024). In Figure 4, we show the development of validity throughout training, which lasted over 4.5 days on an H100 GPU. Throughout training, we fail to match the validity reported in (Bergmeister et al., 2024). Although the validity estimate is quite noisy, it appears to plateau. We select a model checkpoint at 4.8M steps.

**Protein Dataset.** Model selection on the protein graph dataset is challenging, as the MMD metrics computed during validation are noisy, and generating model samples is time-consuming. We take a structured approach and evaluate model checkpoints at 1-4M training steps using the same validation approach as Bergmeister et al. (2024). Namely, for each graph in the validation set, we generate a corresponding model sample with the same number of nodes. We present the resulting MMD metrics in Table 11. Based on these results, we select the model checkpoint at 2M steps.

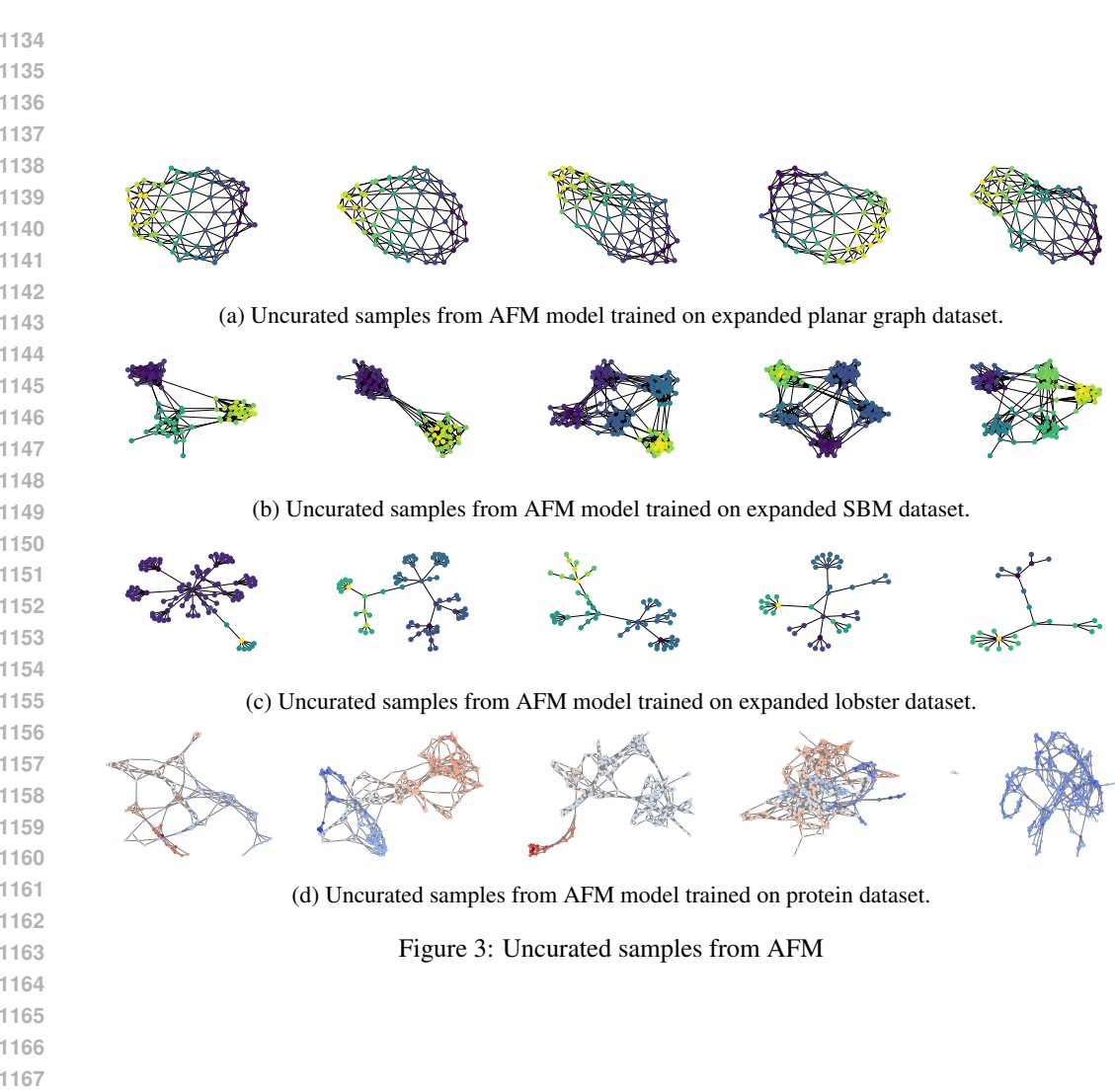

(a) Uncurated samples from AFM model trained on expanded planar graph dataset.

(b) Uncurated samples from AFM model trained on expanded SBM dataset.

(c) Uncurated samples from AFM model trained on expanded lobster dataset.

(d) Uncurated samples from AFM model trained on protein dataset.

Figure 3: Uncurated samples from AFM

Figure 4: SBM validity during training of ESGG on expanded SBM dataset. Validity is computed using 1000 refinement steps in validation but 100 refinement steps during testing to remain consistent with other baselines.

Table 11: Validation results of ESGG trained on protein dataset.

| # Steps | Degree ($\downarrow$) | Clustering ($\downarrow$) | Orbit ($\downarrow$) | Spectral ($\downarrow$) | Wavelet ($\downarrow$) | Ratio ($\downarrow$) |
|---|---|---|---|---|---|---|
| 1M | 0.0242 | 0.1074 | 0.1091 | 0.0095 | 0.0267 | 63.8122 |
| 2M | 0.0028 | 0.0254 | 0.0520 | 0.0009 | 0.0023 | 12.2426 |
| 3M | 0.0066 | 0.0632 | 0.0640 | 0.0030 | 0.0090 | 23.6206 |
| 4M | 0.0293 | 0.1016 | 0.2474 | 0.0079 | 0.0224 | 84.9982 |

## A.12 GRAN HYPERPARAMETERS

For our experiments on the expanded lobster dataset, we use the hyperparameters provided by Liao et al. (2019) for their own (smaller) lobster dataset. For experiments on the expanded planar graph dataset, we utilize the same hyper-parameter setting but reduce the batchsize to 16. For experiments on the SBM dataset, we further reduce the batchsize to 8 and use 2 gradient accumulation steps. For the experiments on the protein dataset, we utilize the pretrained model provided at `http://www.cs.toronto.edu/~rjliao/model/gran_DD.pth`. We perform inference with a batch size of 20.

## A.13 DIGRESS HYPERPARAMETERS

For our experiments on the expanded planar graph and SBM datasets, we use the hyperparameters provided by Vignac et al. (2023) for the corresponding SPECTRE datasets. On the lobster dataset, we use the same hyperparameters as for the expanded SBM dataset (8 layers and batch size 12). On the protein dataset, we use similar hyperparameters as for the expanded SBM dataset but reduce the batch size to 4 due to GPU memory constraints. We use the same inference approach as Vignac et al. (2023), performing generation with a batch size that is twice as large as the batch size used for training. In all cases, we follow Vignac et al. (2023) in using 1000 diffusion steps.

## A.14 ESGG HYPERPARAMETERS

For our experiments on the expanded planar graph and SBM datasets, we use the hyperparameters provided by Bergmeister et al. (2024) for the corresponding SPECTRE datasets. For the expanded lobster dataset, we use the hyperparameters used by Bergmeister et al. (2024) for their tree dataset. We use the test batch sizes provided by Bergmeister et al. (2024) in their hyperparameter configurations.

## A.15 GRAN MODEL SELECTION

**Expanded Planar.** In Table 12, we present validation results of the GRAN model trained on the expanded planar graph dataset. We observe no clear development in model performance past 500 steps. We select the checkpoint at 1000 steps.

Table 12: Validation results for GRAN model trained on expanded planar graph dataset. Evaluated on 260 model samples.

| # Steps | Valid ($\uparrow$) | Node Count ($\downarrow$) | Degree ($\downarrow$) | Clustering ($\downarrow$) | Orbit ($\downarrow$) | Spectral ($\downarrow$) |
|---|---|---|---|---|---|---|
| 500 | 0.00 | 0.0065 | 0.0087 | 0.1749 | 0.0693 | 0.0096 |
| 1000 | 0.00 | 0.0007 | 0.0070 | 0.1696 | 0.1100 | 0.0086 |
| 1500 | 0.77 | 0.0100 | 0.0066 | 0.1730 | 0.0743 | 0.0078 |
| 2000 | 0.00 | 0.0021 | 0.0056 | 0.1658 | 0.0816 | 0.0094 |
| 2500 | 0.77 | 0.0033 | 0.0064 | 0.1768 | 0.1042 | 0.0087 |

**Expanded SBM.** In Table 13, we present validation results of the GRAN model trained on the expanded SBM dataset. We find that, overall, the checkpoint at 200 steps appears to perform best and select it.

Table 13: Validation results for GRAN model trained on expanded SBM dataset. Evaluated on 260 model samples.

| # Steps | Valid ($\uparrow$) | Node Count ($\downarrow$) | Degree ($\downarrow$) | Clustering ($\downarrow$) | Orbit ($\downarrow$) | Spectral ($\downarrow$) |
|---|---|---|---|---|---|---|
| 100 | 22.31 | 1.9992 | 0.0243 | 0.0119 | 0.0400 | 0.0037 |
| 200 | 24.23 | 1.9998 | 0.0194 | 0.0114 | 0.0290 | 0.0026 |
| 400 | 20.38 | 1.9999 | 0.0278 | 0.0130 | 0.0448 | 0.0039 |
| 600 | 20.38 | 1.9999 | 0.0225 | 0.0120 | 0.0318 | 0.0030 |

**Expanded Lobster.** In Table 14, we present validation results of the GRAN model trained on the expanded lobster dataset. We observe no improvement in validity past 2500 steps and select this checkpoint.

Table 14: Validation results for GRAN model trained on expanded lobster graph dataset. Evaluated on 260 model samples.

| # Steps | Valid ($\uparrow$) | Node Count ($\downarrow$) | Degree ($\downarrow$) | Clustering ($\downarrow$) | Orbit ($\downarrow$) | Spectral ($\downarrow$) |
|---|---|---|---|---|---|---|
| 500 | 2.34 | 2.0000 | 0.0257 | 0.4753 | 0.2507 | 0.0509 |
| 1500 | 38.67 | 2.0000 | 0.0092 | 0.0112 | 0.1624 | 0.0329 |
| 2500 | 42.58 | 2.0000 | 0.0083 | 0.0059 | 0.1749 | 0.0361 |
| 3500 | 42.97 | 2.0000 | 0.0101 | 0.0049 | 0.1970 | 0.0406 |

### A.16   ADDITIONAL ABLATIONS

**GAN Tuning.** In Tables 15 and 16, we compare models after training stage I and II on the expanded SBM and lobster datasets from Sec. 4.2. Again, we observe that adversarial fine-tuning substantially improves performance in terms of validity and MMD metrics.

Table 15: Performance of AFM models after stage I (200k steps) and stage II on expanded SBM dataset. Showing median $\pm$ maximum deviation across three runs. All models attain perfect uniqueness and novelty scores.

|  | Stage II | Stage I |
|---|---|---|
| VUN ($\uparrow$) | **75.98** $_{\pm\,3.71}$ | 39.65 $_{\pm\,4.69}$ |
| Degree ($\downarrow$) | **0.0014** $_{\pm\,0.0062}$ | 0.0023 $_{\pm\,0.0063}$ |
| Clustering ($\downarrow$) | **0.0051** $_{\pm\,0.0009}$ | 0.0082 $_{\pm\,0.0012}$ |
| Spectral ($\downarrow$) | **0.0011** $_{\pm\,0.0006}$ | 0.0032 $_{\pm\,0.0006}$ |
| Orbit ($\downarrow$) | **0.0180** $_{\pm\,0.0171}$ | 0.0210 $_{\pm\,0.0135}$ |

**Filtration Function.** In Table 17, we study alternative filtration functions. We compare the line fiedler function to the centrality-based filtration functions introduced in Sec. 3.1. Following Anthonisse (1971); Brandes (2008), we let $\sigma(i,j)$ denote the number of shortest paths between two nodes $i,j \in V$, and $\sigma(i,j\,|\,e)$ denote the number of these paths passing through an edge $e \in E$. Then, we define the betweenness centrality function as:

$$f_{\text{between}}(e) := \sum_{i,j \in V} \frac{\sigma(i,j|e)}{\sigma(i,j)}, \qquad \forall\, e \in E. \tag{27}$$

Based on this, we define the remoteness centrality as $f_{\text{remote}}(e) = -f_{\text{between}}(e)$. We observe that the line fiedler function appears to out-perform the two alternatives in our setting.

**Scheduling.** We study the performance of the three schedules (linear, convex, and concave) proposed in Sec. 3.1 on the planar graph dataset in Table 18. We find that no single variant performs

Table 16: Performance of models after stage I (100k steps) and stage II on expanded lobster dataset. Showing median ± maximum deviation across three runs.

|  | Stage II | Stage I |
|---|---|---|
| VUN (↑) | **79.10** ± 7.13 | 31.25 ± 4.69 |
| Degree (↓) | 0.0004 ± 0.0013 | **0.0004** ± 0.0010 |
| Clustering (↓) | $\mathbf{7.89 \times 10^{-5}}$ ± 5.32 × 10⁻⁵ | 0.0136 ± 0.0054 |
| Spectral (↓) | **0.0016** ± 0.0028 | 0.0030 ± 0.0012 |
| Orbit (↓) | **0.0010** ± 0.0156 | 0.0073 ± 0.0026 |
| Unique (↑) | **99.80** ± 0.10 | 99.51 ± 0.39 |
| Novel (↑) | **100.00** ± 0.10 | 99.90 ± 0.39 |

Table 17: Performance after training stage I with different filtration functions for 100k steps on expanded planar graph dataset. Showing median of three runs for spectral variant and one run each for betweenness and remoteness variants.

|  | VUN (↑) | Degree (↓) | Clustering (↓) | Spectral (↓) | Orbit (↓) |
|---|---|---|---|---|---|
| Line Fiedler | **20.21** | **0.0058** | **0.1768** | **0.0048** | **0.0129** |
| Betweenness | 0.20 | 0.0069 | 0.2724 | 0.0124 | 0.0804 |
| Remoteness | 3.52 | 0.0136 | 0.2720 | 0.0085 | 0.0234 |

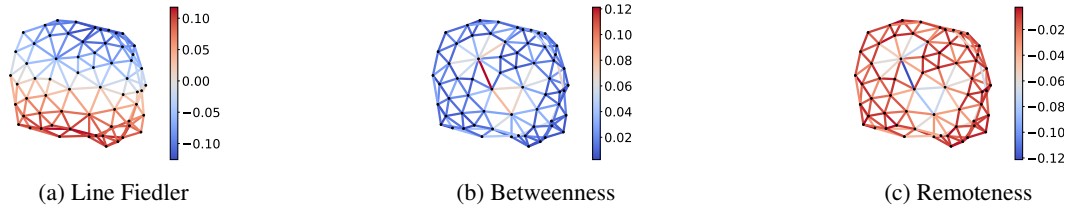

(a) Line Fiedler      (b) Betweenness      (c) Remoteness

Figure 5: Visualization of different filtration functions on a planar graph

Table 18: Performance after training stage I with different filtration schedules for 100k steps on expanded planar graph dataset. All models attain perfect uniqueness and novelty scores. Showing median of three runs for linear variant and one run each for convex and concave variants.

|  | VUN (↑) | Degree (↓) | Clustering (↓) | Spectral (↓) | Orbit (↓) |
|---|---|---|---|---|---|
| Linear | 20.21 | 0.0058 | 0.1768 | 0.0048 | 0.0129 |
| Convex | 5.66 | **0.0043** | 0.2239 | **0.0040** | **0.0062** |
| Concave | **31.05** | 0.0045 | **0.1590** | 0.0059 | 0.0153 |

consistently best across all evaluation metrics. However, the concave variant attains the highest validity score.

**Node Individualization.** In Table 19, we study different node individualization techniques on the expanded planar graph dataset. We refer to the ordering scheme we describe in Sec. 3.2 as the *derived ordering*, as it is based on the filtration function. Additionally, we study individualizations via either *random orderings* or i.i.d. *gaussian noise* that is re-applied in each time-step. Finally, we also consider a variant in which no individualization is applied, i.e., the embedding matrix $W^{\mathrm{node}}$ is fixed to be all-*zeros*. We find that individualizing nodes with learned embeddings based on some

Table 19: Performance after training stage I with different node individualization techniques for 100k steps on expanded planar graph dataset. All models attain perfect uniqueness and novelty scores. Showing median of three runs for derived ordering and one run each for all other variants.

|  | VUN ($\uparrow$) | Degree ($\downarrow$) | Clustering ($\downarrow$) | Spectral ($\downarrow$) | Orbit ($\downarrow$) |
|---|---|---|---|---|---|
| Derived Ordering | **20.21** | 0.0058 | **0.1768** | 0.0048 | 0.0129 |
| Random Ordering | 18.36 | 0.0085 | 0.2332 | **0.0023** | 0.0091 |
| Gaussian Noise | 12.99 | 0.0086 | 0.2356 | 0.0031 | 0.0112 |
| Zeros | 13.48 | **0.0057** | 0.2195 | 0.0023 | **0.0091** |

ordering (either random or derived from the filtration functions) appears to be beneficial. On the planar graph dataset, there is no clear benefit of the derived ordering over random orderings. However, we observe a clear advantage on the SBM dataset, as can be seen in Table 20.

Table 20: Performance after training stage I with derived and random node ordering after 100k steps on expanded SBM datasets. Showing median $\pm$ maximum deviation across three runs for derived ordering and one run for random ordering.

|  | Derived Ordering | Random Ordering |
|---|---|---|
| VUN ($\uparrow$) | **26.95** $\pm$ 2.64 | 2.44 |
| Degree ($\downarrow$) | **0.0222** $\pm$ 0.0127 | 0.0396 |
| Clustering ($\downarrow$) | **0.0106** $\pm$ 0.0012 | 0.0122 |
| Spectral ($\downarrow$) | **0.0061** $\pm$ 0.0014 | 0.0144 |
| Orbit ($\downarrow$) | **0.0548** $\pm$ 0.0244 | 0.0596 |
| Unique ($\uparrow$) | **1.0000** $\pm$ 0.0000 | 0.9951 |
| Novel ($\uparrow$) | **1.0000** $\pm$ 0.0000 | **1.0000** |

**Stage I.** While the ablation study in Sec. 4.4 demonstrates that stage II training substantially boosts performance, we now show that, similarly, stage I is crucial too. To this end, we perform stage II training on a very early checkpoint of stage I training. Specifically, we use a checkpoint obtained after 10k steps of stage I training on the expanded planar graph dataset. We present the results in Table 21. We observe that performing stage II training on a premature checkpoint from stage I substantially harms performance. Hence, stage I training is a crucial part of our method.

Table 21: Performance on expanded planar graph dataset of AFM variants with different training durations during stage I. Showing median across three runs for 200k steps and a single run for 10k steps.

| # Stage I Steps | VUN ($\uparrow$) | Deg. ($\downarrow$) | Clus. ($\downarrow$) | Orbit ($\downarrow$) | Spec. ($\downarrow$) |
|---|---|---|---|---|---|
| 200k | **79.20** | **0.0004** | **0.0183** | **0.0002** | **0.0012** |
| 10k | 3.32 | 0.0016 | 0.2278 | 0.0464 | 0.0069 |

## A.17 BIAS AND VARIANCE OF ESTIMATORS

Previous works (Martinkus et al., 2022; Vignac et al., 2023; Bergmeister et al., 2024) evaluate their graph generative models on as few as 40 samples. In this section, we investigate how this practice impacts the variance and bias of the estimators used in model evaluation and argue that a higher number of test samples should be chosen.

### A.17.1 VARIANCE OF VALIDITY ESTIMATION

On synthetic datasets such as those introduced in (Martinkus et al., 2022), one may verify whether model samples are "valid", i.e., whether they satisfy a property that is fulfilled by (almost) all samples of the true data distribution. By taking the ratio of valid graphs out of $n$ model samples, previous works have estimated the probability of obtaining valid graphs from the generator.

**Definition 1.** *Let the random variable $G$ denote a sample from a graph generative model and let* valid $: \mathcal{G} \to \{0, 1\}$ *a measurable binary function that determines whether a sample is valid. Then the models true validity ratio is defined as:*

$$\mathbb{P}[\text{valid}(G) = 1] \tag{28}$$

*For i.i.d. samples $G_1, \ldots, G_n$, we introduce the following estimator:*

$$V := \frac{\sum_{i=1}^n \text{valid}(G_i)}{n} \tag{29}$$

Given the simplicity of the validity metric, we can very easily derive the uncertainty of the estimator used for evaluation. We make this concrete in Proposition 5.

**Proposition 5.** *For a generative model with a true validity ratio of $p \in [0, 1]$, the validity estimator on $n$ samples is unbiased and has standard deviation $\sqrt{p(1-p)}/\sqrt{n}$.*

*Proof.* Assuming that the random variables $G_1, \ldots, G_n$ are i.i.d. samples from the generative model, then the random variables $\text{valid}(G_1), \ldots, \text{valid}(G_n)$ are i.i.d. according to $\text{Bernoulli}(p)$. The validity estimator is given as:

$$V = \frac{\sum_{i=1}^n \text{valid}(G_i)}{n} \tag{30}$$

By the linearity of expectation, we have

$$\mathbb{E}[V] = \frac{\sum_{i=1}^n \mathbb{E}[\text{valid}(G_i)]}{n} = \frac{np}{p} = p \tag{31}$$

which shows that the estimator is unbiased. The variance is given by:

$$\begin{aligned}
\text{Var}[V] &= \frac{\text{Var}\left[\sum_{i=1}^n \text{valid}(G_i)\right]}{n^2} = \frac{\sum_{i=1}^n \text{Var}[\text{valid}(G_i)]}{n^2} \\
&= \frac{p(1-p)}{n}
\end{aligned} \tag{32}$$

where we used the independence assumption in the first line. Taking the square root, we obtain the standard deviation from the proposition. □

From Proposition 5, we note that the standard deviation of the validity estimate can be as high as $1/(2\sqrt{n})$, which is achieved at $p = 0.5$. For $n = 40$, we find that the standard deviation can therefore be as high as 7.9 percentage points.

### A.17.2 BIAS AND VARIANCE OF MMD ESTIMATION

**Definition 2.** *Let $(\mathcal{X}, d)$ be a metric space and let $k : \mathcal{X} \times \mathcal{X} \to \mathbb{R}$ be a measurable, symmetric kernel which is bounded but not necessarily positive-definite. Let $X := [x_1, \ldots, x_n]$ be i.i.d. samples from a Borel distribution $p_x$ on $\mathcal{X}$ and $Y := [y_1, \ldots, y_n]$ be i.i.d. samples from a distribution*

$p_y$. Assume $X$ and $Y$ to be independent. Following (Gretton et al., 2012), define the squared MMD of $p_x$ and $p_y$ as:

$$\text{MMD}^2(p_x, p_y) := \mathbb{E}[k(x_1, x_2)] + \mathbb{E}[k(y_1, y_2)] - 2\mathbb{E}[k(x_1, y_1)] \tag{33}$$

and note that this is well-defined by our assumptions. Finally, introduce the following estimator for the squared MMD:

$$M := \frac{1}{n^2} \sum_{i,j=1}^{n} k(x_i, x_j) + \frac{1}{m^2} \sum_{i,j=1}^{m} k(y_i, y_j) - \frac{2}{nm} \sum_{i=1}^{n} \sum_{j=1}^{m} k(x_i, y_j) \tag{34}$$

We empirically study bias and variance of the MMD estimates on the planar graph dataset. We generate 8192 samples from one of our trained model and repeatedly compute the MMD between the test set and a random subset of those samples. We vary the size of the random subsets and run 64 evaluations for each size, computing mean and standard deviation of the MMD metrics across the 64 evaluations. We report the results in Table 22. We observe that on average the MMD is severly

Table 22: Mean MMD $\pm$ standard deviation across 64 evaluation runs of a single model. The test set contains 256 planar graphs, while a varying number of model samples is used, as indicated on the left. The MMD and its variance decrease substantially with larger numbers of model samples.

| # Model Samples | Degree ($\downarrow$) | Clustering ($\downarrow$) | Spectral ($\downarrow$) |
|---|---|---|---|
| 32 | $8.59 \times 10^{-4} \pm {}_{5.59 \times 10^{-4}}$ | $4.21 \times 10^{-2} \pm {}_{1.44 \times 10^{-2}}$ | $4.73 \times 10^{-3} \pm {}_{9.14 \times 10^{-4}}$ |
| 64 | $5.58 \times 10^{-4} \pm {}_{2.90 \times 10^{-4}}$ | $2.68 \times 10^{-2} \pm {}_{7.58 \times 10^{-3}}$ | $2.59 \times 10^{-3} \pm {}_{4.78 \times 10^{-4}}$ |
| 128 | $4.40 \times 10^{-4} \pm {}_{1.79 \times 10^{-4}}$ | $2.17 \times 10^{-2} \pm {}_{4.33 \times 10^{-3}}$ | $1.61 \times 10^{-3} \pm {}_{3.16 \times 10^{-4}}$ |
| 256 | $4.39 \times 10^{-4} \pm {}_{1.45 \times 10^{-4}}$ | $2.02 \times 10^{-2} \pm {}_{3.89 \times 10^{-3}}$ | $1.14 \times 10^{-3} \pm {}_{1.86 \times 10^{-4}}$ |
| 512 | $4.32 \times 10^{-4} \pm {}_{8.48 \times 10^{-5}}$ | $1.81 \times 10^{-2} \pm {}_{2.56 \times 10^{-3}}$ | $1.18 \times 10^{-3} \pm {}_{2.04 \times 10^{-4}}$ |
| 1024 | $4.26 \times 10^{-4} \pm {}_{5.99 \times 10^{-5}}$ | $1.72 \times 10^{-2} \pm {}_{1.66 \times 10^{-3}}$ | $1.18 \times 10^{-3} \pm {}_{2.00 \times 10^{-4}}$ |

over-estimated when using fewer than 256 model samples. At the same time, the variance between evaluation runs is large when few samples are used, making the results unreliable.

### A.18 ADVERSARIAL FINETUNING DETAILS

We provide pseudocode for the adversarial fine-tuning stage in Algorithm 1. We note that we do not make all procedures explicit and that many hyper-parameters must be chosen (including the number of steps and epochs in TRAINGENERATORANDVALUEMODEL).

**Generator.** The generator operates in inference mode, meaning that all dropout layers are disabled and batch normalization modules utilize the (now frozen) moving averages from training stage I. Hence, the behavior of the generative model becomes reproducible. It acts as a stochastic policy in a higher-order MDP, where the graphs $G_0, \ldots, G_T$ are the states. It receives a terminal reward for the plausibility of the final sample $G_T$.

**Discriminator.** The discriminator is implemented as a GraphGPS (Rampásek et al., 2022) model which performs binary classification on graph samples $G_T$, distinguishing real samples from generated samples. It is trained via binary cross-entropy on batches consisting in equal proportions of generated graphs and graphs from the dataset $\mathcal{D}$. For a given graph $G_T$, the discriminator produces a probability of "realness" by applying the sigmoid function to its logit. Following SeqGAN (Yu et al., 2017), the log-sigmoid of the logit then acts as a terminal reward for the generative model. We emphasize that only the final graph $G_T$ is presented to the discriminator.

**Value Model.** The value model uses the same backbone architecture as our generative model and regresses scalars from pooled node representations. It is trained via least squares regression. The value model is used to compute baselined reward-to-go values.

**Training Outline.** While Algorithm 1 provides a technical description of the training algorithm, we also provide a rougher outline here. At the start of training stage II, the generator is initialized with the weights learned in training stage I, while the discriminator and value model are initialized randomly. Before entering the main training loop, we pre-train discriminator and value model to match the generator. Namely, we first pre-train the discriminator to classify graphs as either "real" or "generated". The log-likelihood of "realness" acts as a terminal reward of the generative model. The discriminator is then pre-trained to regress the reward-to-go. After pre-training is finished, we proceed to the training loop, which consists of alternating training of (i) the generator and value model and (ii) the discriminator. As described above, the generator is trained via PPO to maximize the terminal reward provided by the discriminator. The value model is used to baseline the reward and is continuously trained to regress the reward-to-go. The discriminator, on the other hand, continues to be trained on generated and real graph samples via binary cross-entropy.

---

**Algorithm 1** Adversarial Finetuning

---

**procedure** TRAINGENERATORANDVALUEMODEL($p_\theta, d_\varphi, v_\vartheta$)
    **for** $i = 1 \ldots, N_{\text{steps}}$ **do**
        $\mathcal{S} \leftarrow [\,]$                                      ▷ List of sampled filtrations
        $r \leftarrow 0 \in \mathbb{R}^{N_{\text{samples}}}$                           ▷ Terminal rewards
        **for** $j = 1 \ldots, N_{\text{samples}}$ **do**
            $G_0^{(j)}, \ldots, G_T^{(j)} \leftarrow$ SAMPLEFILTRATION($p_\theta$)
            $\mathcal{S} . \text{append}\left(\left(G_0^{(j)}, \ldots, G_T^{(j)}\right)\right)$
            $r_j \leftarrow \text{logsigmoid}(d_\varphi(G_T^{(j)}))$
            $r_j \leftarrow \max(r_j, R_{\text{lower}})$                     ▷ Reward clamping
        **end for**
        $r \leftarrow$ WHITEN($r$)                       ▷ Whiten rewards using EMA of mean and std
        $g_{j,t} \leftarrow 0 \qquad \forall j = 1, \ldots, N_{\text{samples}} \; \forall t = 0, \ldots, T-1$         ▷ Rewards-to-go
        **for** $j = 1 \ldots, N_{\text{samples}}$ **do**
            **for** $t = 0, \ldots, T-1$ **do**
                $g_{j,t} \leftarrow r_j - v_\vartheta(G_0^{(j)}, \ldots, G_t^{(j)})$         ▷ Compute baselined RTG
            **end for**
        **end for**
        TRAINVALUEMODEL($v_\vartheta, \mathcal{S}, r$)
        **for** $k = 1 \ldots, N_{\text{epoch}}$ **do**
            $l_{j,t}^{(k)} \leftarrow -\log p_\theta(G_t^{(j)} | G_{t-1}^{(j)}, \ldots, G_0^{(j)}) \qquad \forall j = 1, \ldots, N_{\text{samples}} \quad \forall t = 1, \ldots, T$
            $u_{j,t} \leftarrow \exp(\text{sg}[l_{j,t}^{(1)}] - l_{j,t}^{(k)}) \qquad \forall j, t$
            $\mathcal{L}_{j,t}^{(1)} \leftarrow -u_{j,t} \cdot g_{j,t-1} \qquad \forall j, t$
            $\mathcal{L}_{j,t}^{(2)} \leftarrow -\text{clamp}(u_{j,t}, 1-\epsilon, 1+\epsilon) \cdot g_{j,t-1} \qquad \forall j, t$
            $\mathcal{L} \leftarrow \sum_{j,t} \max(\mathcal{L}_{j,t}^{(1)}, \mathcal{L}_{j,t}^{(2)})$
            $\theta \leftarrow \theta - \delta \nabla_\theta \mathcal{L}$                     ▷ Backpropagate and update parameters
        **end for**
    **end for**
**end procedure**

**procedure** GANTUNING($p_\theta, \mathcal{D}$)         ▷ Takes generator from training stage I and graph dataset
    $d_\varphi \leftarrow$ new GNN                                ▷ Initialize discriminator
    TRAINDISCRIMINATOR($p_\theta, d_\varphi, \mathcal{D}$)                     ▷ Pre-train discriminator
    $v_\vartheta \leftarrow$ new mixer model
    $\mathcal{S} \leftarrow$ GENERATEFILTRATIONS($p_\theta$)
    $r \leftarrow$ GRADESAMPLES($\mathcal{S}, d_\varphi$)
    TRAINVALUEMODEL($v_\vartheta, \mathcal{S}, r$)                        ▷ Pre-train value model
    **while** not converged **do**
        TRAINGENERATORANDVALUEMODEL($p_\theta, d_\varphi, v_\vartheta$)
        TRAINDISCRIMINATOR($p_\theta, d_\varphi, \mathcal{D}$)
    **end while**
**end procedure**

---

