# OpenReview forum: "Towards Fast Graph Generation via Autoregressive Filtration Modeling"
_ICLR.cc/2025/Conference — Submitted to ICLR 2025_

### Official Review · Reviewer_xUTE · 2024-10-27

**Soundness:** 3
**Presentation:** 3
**Contribution:** 3
**Rating:** 8
**Confidence:** 4

**Summary:**

This paper proposes AFM, a methodology to use filtration sequences to generate graphs autoregressively using parts of subgraphs. Compared with existing methods, the proposed AFM achieves much faster graph generation with similar and competitive performance.

**Strengths:**

1. The paper is well-written, easy to follow and understand. Specifically, the related work section is well-structured, covering the most relevant and important research works in graph generation.

2. The authors tried three different schedule functions: linear, convex, and concave, to better control the graph generation process. Experiments with different schedule functions showcase that each can perform better than others in terms of specific metrics.

3. The running speed of the proposed AFM method is impressive, with competitive performance in terms of various metrics compared to other baselines.

4. The ablation study showcases the importance of noise augmentation.

**Weaknesses:**

1. The authors made the assumption that only connected graphs are presented in the model during training. However, I am slightly concerned that this assumption might be too strong, which could limit the applicability of the proposed AFM method in generating certain real-world graphs.

2.The baselines that the authors compared with in Section 4.2 (expanded synthetic datasets) and Section 4.3 (real-world datasets) are too few. Comparison with more baselines could better showcase the proposed method.

**Questions:**

1. For the filtration function, the authors chose the line Fiedler function $f_{Fiedler}$, which is the second smallest eigenvector of the symmetrically normalized Laplacian of the line graph $L(G)$. Is there any good theoretical or at least intuitive explanation as to why this $f_{Fiedler}$ might perform well?

2. I am a bit curious about the scalability of the proposed method in terms of the size of the graph it can generate. How does the runtime of the AFM method change as the graph size N increases?

---

> ### Author Response · Authors · 2024-11-24
> **Author Response to Reviewer xUTE**
>
> Thank you for your time and effort in providing such constructive feedback. We greatly appreciate your insights and have addressed the points you raised below.
>
> ## Weaknesses
> > The authors made the assumption that only connected graphs are presented in the model during training
>
> This assumption was made to ensure that the line graph of $G_T$ is connnected. Then, eigenvectors corresponding to the second smallest eigenvalue (i.e. line Fiedler vectors) will paint the edges of $G_T$ in a continuous gradient. If $G_T$ is disconnected, these eigenvectors may paint edges of only a single component in a continous gradient while edges of other components are assigned the constant value 0. Thus, without modification of the filtration function, AFM would have to generate whole connected components in a single generation step.
>
> To rectify this and extend AFM to disconnected graphs, one has to adjust the line Fiedler filtration function. One simple approach would be defining the line Fiedler function separately per connected component.
>
> Hence, the reason for this assumption was technical and modifications of the filtration function should allow applications on disconnected graphs.
>
>
> > The baselines that the authors compared with in Section 4.2 (expanded synthetic datasets) and Section 4.3 (real-world datasets) are too few.
>
> We have added two baselines from the literature to Section 4.1. Unfortunately, due to limited computational resources and the strict timeline, we do not expect that we will be able to run additional baselines on our own datasets. However, we believe that our current evaluations already adequately demonstrate the quality and efficiency of our proposed model: AFM outperforms state-of-the-art autoregressive models, achieving generation quality comparable to that of diffusion baselines while being substantially faster during inference.
>
> ## Questions
> > Is there any good theoretical or at least intuitive explanation as to why [the Fiedler vector of the line graph]  might perform well?
>
> Ultimately, the line Fiedler filtration function is a heuristic choice and we don't have a rigorous explanation as to why it works well. Intuitively, the line Fiedler function paints edges of a graph in a "continuous gradient", assigning similar values to neighboring edges. We hypothesize that this may simplify the task of learning to reverse the filtration, as edges are mostly added in locally confined regions that neighbor the edges that were added in the previous steps.
>
> To guide your intuition, the illustration of the behavior of the line Fiedler filtration in the appendix may be helpful (Figure 5 titled "Visualization of different filtration functions on a planar graph").
>
> > How does the runtime of the AFM method change as the graph size N increases?
>
> As noted in the common response, we have added a theoretical runtime analysis to the main text and an investigation of a first-order autoregressive variant to the appendix.

---

### Official Review · Reviewer_JXGh · 2024-11-08

**Soundness:** 2
**Presentation:** 3
**Contribution:** 2
**Rating:** 6
**Confidence:** 4

**Summary:**

The paper introduces Autoregressive Filtration Modeling (AFM), a novel approach to graph generation that leverages the concept of graph filtration from topological data analysis. AFM transforms a graph into a sequence of nested subgraphs, enabling a structured and efficient autoregressive graph generation process. The method integrates noise augmentation and adversarial fine-tuning with reinforcement learning to address challenges like exposure bias.

**Strengths:**

- The application of graph filtration for graph generation is a novel and unique contribution, moving beyond existing approaches like diffusion models and node/edge-addition methods.
- The integration of reinforcement learning for adversarial fine-tuning further distinguishes AFM from other autoregressive methods.

**Weaknesses:**

- The multi-step approach (including filtration, autoregressive modeling, noise augmentation, and adversarial fine-tuning) may be challenging to implement and tune, particularly for practitioners.
- The focus on non-attributed graphs limits the model’s applicability to scenarios involving node or edge labels, a common requirement in many practical applications.
- The paper can be better motivated by stating why there is a need for graph generation algorithms especially for high-throughput applications. And also why is graph generation important. These are missing in the introduction.
- The experimental results, among all, do not stand out the superiority of the proposed method against baselines.

**Questions:**

- How could AFM be extended to handle node or edge attributes? Are there plans to address this limitation in future work?
- How to extend AFM to work on directed graphs?
- Can we solve the problem of high computational complexity of previous autoregressive and diffusion models with distributed computing resources?
- Please justify why diversity and structural consistency are important for filtration function?
- What is $f_\text{remote}$ in line 200?
- What does 'appropriate edges' in line 292 refer to?

---

> ### Author Response · Authors · 2024-11-24
> **Author Response to Reviewer JXGh (1/2)**
>
> Thank you for your time and effort in providing such constructive feedback. We greatly appreciate your insights and have addressed the points you raised below.
>
> ## Weaknesses
> > The multi-step approach (including filtration, autoregressive modeling, noise augmentation, and adversarial fine-tuning) may be challenging to implement and tune, particularly for practitioners.
>
> We agree that our method is technically complex. To facilitate adoption and ease of use, we provide additional practical advice on tuning hyper-parameters in the appendix of our revised paper (please see "A.5 Practical Advice on Hyperparameter Choice"). Additionally, we will publish our source code upon acceptance of the paper.
>
> > The focus on non-attributed graphs limits the model’s applicability to scenarios involving node or edge labels, a common requirement in many practical applications.
>
> We acknowledge the concern that our current design does not explicitly support graphs with node and edge attributes. However, we want to emphasize that our primary focus is on developing efficient AR model for unattributed graphs and we present the **first AR model that matches the performance of diffusion-based models while maintaining its speed advantage**. Furthermore, un-attributed graph topologies are often of significant interest in various applications, such as protein structure modeling and design, where contact maps represent the spatial proximity between amino acids [1, 2].
>
> As discussed in our conclusion and appendix, our method is naturally extendable to edge-labeled graphs. Regarding the modeling of node attributes, we propose two prospective approaches:
> - Post-hoc node labeling: After generating the graph topology, node labels can be assigned based on the structural context, as suggested by Bergmeister et al. (2024). For example, in molecular graphs, valency constraints and local connectivity patterns may provide strong cues for determining atom types. Such an approach was used in [3].
> - Extension of the Filtration Process: We can extend the filtration process to include node labels by representing them as labeled self-loops and incorporating them into the edge filtration. This approach allows the model to learn node attributes alongside the graph topology within the same generative framework.
>
> We believe that integrating node and edge labels is a promising direction for future work, which can further enhance the applicability of our approach to a broader range of graph generation tasks.
>
>
> > The paper can be better motivated by stating why there is a need for graph generation algorithms especially for high-throughput applications
>
> We have updated the introduction to highlight this point. Previous works have demonstrated that the spaces of drug-like molecules and protein conformations are practically infinte. Hence, high-throughput generation is emerging as a promising alternative to screening of pre-defined graph libraries.
>
> > The experimental results, among all, do not stand out the superiority of the proposed method against baselines.
>
> We have demonstrated that our method achieves generation quality that is comparable to the quality achieved by state-of-the-art diffusion models while being **two orders of magnitude faster**. We do not aim to out-perform existing diffusion baselines in terms of quality, but aim to substantially improve efficiency without compromising on quality.
> **To the best of our knowledge, our method is the first autoregressive graph generative model that can compete with diffusion models on the benchmarks we presented.**
>
> ## References
> [1] Zeming Lin et al., Evolutionary-scale prediction of atomic-level protein structure with a language model. Science379, 1123-1130(2023). DOI:10.1126/science.ade2574
>
> [2] Chen, Y., Xu, Y., Liu, D. et al. An end-to-end framework for the prediction of protein structure and fitness from single sequence. Nat Commun 15, 7400 (2024). https://doi.org/10.1038/s41467-024-51776-x
>
> [3] Li, Yibo, Jianxing Hu, Yanxing Wang, Jielong Zhou, Liangren Zhang, and Zhenming Liu. ‘DeepScaffold: A Comprehensive Tool for Scaffold-Based De Novo Drug Discovery Using Deep Learning’. Journal of Chemical Information and Modeling 60, no. 1 (2020): 77–91. https://doi.org/10.1021/acs.jcim.9b00727.

---

> ### Author Response · Authors · 2024-11-24
> **Author Response to Reviewer JXGh (2/2)**
>
> ## Questions
> > How could AFM be extended to handle node or edge attributes?
>
> See the discussion above.
>
> > How to extend AFM to work on directed graphs?
>
> While we do not present experiments on directed graph datasets, our presented method should naturally extend to directed graphs. We have updated the appendix ("A.7 Edge Decoder Architecture") to elaborate on this.
>
> > Can we solve the problem of high computational complexity of previous autoregressive and diffusion models with distributed computing resources?
>
> Denoising diffusion models like DiGress are inherently sequential with a large number of denoising iterations, making parallelization challenging. Distributed computing resources can be used for data parallelism (e.g. generating large numbers of graphs). While this will surely increase throughput, it will naturally lead to proportionally large increases in computational costs. In our opinion, the need for efficient graph generative models remains.
>
> > Please justify why diversity and structural consistency are important for filtration function?
>
> - Filtration functions that do not assign distinct values to different edges (i.e. are not diverse) may lead to graph filtrations in which large numbers of edges are added in a single step, regardless of the choice of $T$. Hence, one cannot steer the granularity of the filtration via the number of steps $T$.
> - By "structural consistency", we refer to the notion that the ordering of edges imposed by the filtration function should carry some information on their relations to each other / their position in the graph. The line Fiedler function, for instance, tends to assign similar values to edges that are close to each other. We hypothesize that this simplifies the task of reversing the filtration.
>
> > What is in line 200?
>
> In this line, we introduce alternative filtration functions based on the concept of betweenness. We provide a rigorous definition in the appendix (see "A.16 Additional Ablations"). We have added a reference to this appendix to improve clarity.
>
> > What does 'appropriate edges' in line 292 refer to?
>
> By "appropriate edges", we refer to edges that are erroneously added in intermediate graphs $G_t$ with $t < T$. For instance, when generating trees, these could be edges that are part of a cycle. When generating planar graphs, these could be edges that are part of a $K_5$ or $K_{3, 3}$ sub-division. In order to obtain a valid graph sample $G_T$ (i.e., one that is cycle-free or planar), some of these edges must be deleted in subsequent steps.

---

> > ### Comment · Reviewer_JXGh · 2024-11-29
> >
> > Dear Authors,
> >
> > Many thanks for the rebuttal and the revision. I am happy that my concerns are addressed. The revision with a better motivation definitely helps the paper to stand out. I will raise the score to 6.
> >
> > Warm regards.

---

> > > ### Author Response · Authors · 2024-11-29
> > >
> > > Thank you for your valuable feedback and support! If you have any further questions or comments, we remain available for discussion.

---

### Official Review · Reviewer_vHcj · 2024-11-08

**Soundness:** 3
**Presentation:** 3
**Contribution:** 3
**Rating:** 6
**Confidence:** 4

**Summary:**

The paper introduces Autoregressive Filtration Modeling (AFM), a new method for graph generation. AFM transforms graphs into sequences of monotonically increasing subgraphs. It incorporates an autoregressive graph mixer, noise augmentation to reduce exposure bias, and reinforcement learning for refining the model.

**Strengths:**

S1: Improving the efficiency of graph generation is an interesting research direction of practical importance.

S2: The proposed mixture-of-Bernoullis component enhances the expressiveness of the generative model.

S3: The proposed adversarial finetuning method significantly boosts the fidelity of generated graphs.

S4: The authors clearly disclosed experimental details to facilitate reproducibility.

**Weaknesses:**

W1: The motivation is inconsistent with the proposed method. Section 3.1 states that a filtration has G_t ⊇ G_{t-1}. However, the proposed method in Section 3.2 both add and delete edges, which violates the definition of a filtration. It is more like a diffusion model except that they replace p(G_t|G_{t-1}) with p(G_t|G_{t-1},...,G_0). The objective function looks the same as that of DiGress (a diffusion model). Hence, it might make more sense if the authors motivate the proposed method from a diffusion model perspective instead of a filtration perspective.

W2: The experimental results are not presented in a very meaningful manner. It seems that the speedup comes mainly from the small number of steps (i.e., T) of the proposed method AFM. For example, on the Expanded Lobster dataset, DiGress runs 1000 steps for 4.86 sec and achieves VUN=96.58 while the proposed AFM runs 30 steps for 0.03 sec but achieves only VUN=79.10. Thus, directly comparing them does not provide much information. To provide a more meaningful comparison, the authors can plot a VUN-vs-time curve for each method (by varying T) and compare these curves instead.

W3: The time complexity of autoregressive generation is O(T^2). Thus, it might be inefficient if we seek to increase the generation quality by increasing T.

W4: The ablation studies are a bit insufficient. The authors did not report the performence of using only stage II (i.e., no stage I). Thus, it is unclear how effective the proposed stage I is. Besides that, the authors did not provide ablation studies on performance vs K.

W5: The comparison is a bit unfair. As the proposed adversarial finetuning (AF) is orthogonal to the generative model, comparing the proposed method with AF and baseline methods without AF is not an apple-to-apple comparison.

**Questions:**

See weaknesses.

---

> ### Author Response · Authors · 2024-11-24
> **Author Response to Reviewer vHcj**
>
> Thank you for your time and effort in providing such constructive feedback. We greatly appreciate your insights and have addressed the points you raised below.
>
> > W1: The motivation is inconsistent with the proposed method. Section 3.1 states that a filtration has G_t ⊇ G_{t-1}. However, the proposed method in Section 3.2 both add and delete edges
>
> While we do use noise-augmented filtrations in practice, they are still largely guided by the original monotonic filtration sequence. Indeed, for a given graph $G_t$ in the (monotonic) filtration sequence, we randomly perturb the presence of edges for at most 25% of all node pairs (in particular, we delete on average at most 25% of edges in $G_t$). The frequency of perturbations is governed by the noise schedule $\lambda(t)$, which is annealed from 25% at $t=1$ to 5% at $t=T - 1$.
>
> We are happy to adjust nomenclature or phrasing if you have suggestions to improve clarity.
>
> > W1: It is more like a diffusion model except that they replace $p(G_t|G_{t-1})$ with $p(G_t|G_{t-1},...,G_0)$. The objective function looks the same as that of DiGress.
>
> We agree that our method resembles diffusion in some aspects: Namely, we also aim to train a model to reverse a corruption process. However, our approach does not fit the formalism of denoising diffusion:
> - In general, we do not restrict oursevels to a first-order autoregressive structure
> - The corruption process (i.e. the noisy filtration) is in general not Markov
> - Our training objective differs from the loss of DiGress.
>
> We have added a discussion of this point to the extended related work section in the appendix.
>
> Our loss resembles the DiGress loss in that we consider a cross-entropy. However, we note that DiGress uses a simplified diffusion loss under an "$x_0$-parametrization" ([see here](https://proceedings.neurips.cc/paper/2021/file/958c530554f78bcd8e97125b70e6973d-Paper.pdf)). Given a sequence of increasingly corrupted graphs $G_T, \dots, G_0$, their loss at timestep $t$ would be $-\log p_\theta(G_T | G_t)$ while our loss would be $-\log p_\theta(G_{t+1} | G_t, ..., G_0)$. I.e., we form a cross-entropy against $G_{t+1}$ while DiGress forms a cross-entropy against the "clean graph" $G_T$ (note that literature on diffusion models uses a reverse indexing for the sequence of graphs $G_t$).
>
> While our loss bears more similarity to the VLB of DiGress (as opposed to the simplified diffusion loss), we maintain that our method does not fit into the formalism of denoising diffusion, as explained above.
>
> > W2: It seems that the speedup comes mainly from the small number of steps (i.e., T) of the proposed method AFM. [...] To provide a more meaningful comparison, the authors can plot a VUN-vs-time curve for each method (by varying T)
>
> We agree that the small number of steps is the main contributor to the speedup compared to other methods. Following your suggestion, we have added this experiment to the main paper (see Figure 2). We vary $T$ both in AFM and DiGress and study the effect on quality and speed. We refer to the section "Effect of filtration granularity $T$" of the general response for a discussion of our findings.
>
> > W3: The time complexity of autoregressive generation is O(T^2). Thus, it might be inefficient if we seek to increase the generation quality by increasing T.
>
> Please refer to the "Complexity Analysis" section in our general response.
>
> > W4: The ablation studies are a bit insufficient. The authors did not report the performence of using only stage II
>
> Thank you for this interesting suggestion. We have added an ablation using stage II on a premature checkpoint of stage I ("A.16 Additional Ablations"). Convergence in stage I is necessary to obtain satisfactory performance in stage II.
>
> > W5: As the proposed adversarial finetuning (AF) is orthogonal to the generative model, comparing the proposed method with AF and baseline methods without AF is not an apple-to-apple comparison.
>
> This is a fair point, as AF could also be applied to our autoregressive baseline (GRAN). However, we make the following two observations:
> - In all of our experiments, AFM is faster during inference than GRAN. In particular on larger protein graphs, AFM is ten times faster. This is not affected by adversarial finetuning.
> - As we demonstrate in the ablation experiment for your previous point (W4), adequate performance after the first training stage is a prerequisite for good performance after adversarial finetuning. After training stage I, we already observe that AFM outperforms GRAN substantially in terms of quality (VUN) on the expanded planar graph dataset (c.f. Table 4, "Stage I w/ Noise"). The same holds true for the expanded SBM dataset (c.f. Table 15, "A.16 Additional Ablations"). AFM without adversarial finetuning performs slightly worse than GRAN on the expanded lobster dataset (c.f. Table 16).
>
> With this being said, we agree that studying the effect of AF on other autoregressive approaches would be interesting.

---

> ### Comment · Reviewer_vHcj · 2024-11-26
> **Follow-up comments (edited)**
>
> Thank you for your thorough response. Below are my further comments about your answers.
>
> > *While we do use noise-augmented filtrations in practice, they are still largely guided by the original monotonic filtration sequence. Indeed, for a given graph $G_t$ in the (monotonic) filtration sequence, we randomly perturb the presence of edges for at most 25% of all node pairs (in particular, we delete on average at most 25% of edges in $G_t$). The frequency of perturbations is governed by the noise schedule $\lambda(t)$, which is annealed from 25% at $t=1$ to 5% at $t=T - 1$.*
>
> Thanks for your elaboration. However, I do not fully agree with this. What you call "noise-augmented filtrations" look more like a diffusion model to me. Diffusion models also have a noise schedule, and the generation process is also governed by the denoising process. The main difference seems to be the loss function. This weakens the motivation of using a filtration instead of a diffusion process.
>
> > *We form a cross-entropy against $G_{t+1}$ while DiGress forms a cross-entropy against the "clean graph" $G_T$.*
>
> Thanks for your clarification. I agree that they do have some differences, but they are still more like variants than essential differences to me because theoretically speaking, they are just different upper bounds on $E[-\log p_\theta(G_T)]$. Why does your loss function perform better than DiGress' loss function? Also, I wonder what the performance will be like (i) when you train DiGress using your loss function and (ii) when you train AFM using DiGress' loss function.
>
> > *We agree that the small number of steps is the main contributor to the speedup compared to other methods. Following your suggestion, we have added this experiment to the main paper (see Figure 2). We vary $T$ both in AFM and DiGress and study the effect on quality and speed. We refer to the section "Effect of filtration granularity $T$" of the general response for a discussion of our findings.*
>
> Thanks for the additional experiments. This clearly shows that AFM achieves a better tradeoff than DiGress. However, this comparison considered only discrete-time models but did not consider continuous-time models such as [1,2]. Continuous-time models can achieve a natural tradeoff between generation quality and generation steps by adjusting the discretization points. For example, Table 9 in [1] shows that their continuous-time model still achieves good performance even when using only 5 steps. How does AFM compare with these continuous-time models in terms of the tradeoff?
>
> - [1] Xu et al. Discrete-state continuous-time diffusion for graph generation. arXiv:2405.11416. NeurIPS, 2024.
> - [2] Siraudin et al. Cometh: A continuous-time discrete-state graph diffusion model. arXiv:2406.06449.
>
> > *This is a fair point, as AF could also be applied to our autoregressive baseline (GRAN)...*
>
> Thanks for your clarification. As adversarial finetuning seems to offer a significant improvement as a standalone module, my point is that the adversarial finetuning *might contribute more* than the proposed filtration framework. If so, this would substantially weaken the motivation of this paper. Thus, to rule out this possibility, a better ablation study is to apply adversarial finetuning to stronger baselines like DiGress.
>
> Thanks.

---

> > ### Author Response · Authors · 2024-11-28
> > **Author Response to Follow Up Comments (1/2)**
> >
> > Thank you for your additional feedback! We have addressed your points below:
> >
> > > What you call "noise-augmented filtrations" look more like a diffusion model to me. Diffusion models also have a noise schedule, and the generation process is also governed by the denoising process.
> >
> > We agree with your observation regarding the similarities between "noise-augmented filtrations" and diffusion models, particularly the shared use of a noise schedule and a denoising process.
> > However, we maintain that our approach diverges from the conventional diffusion framework in significant ways. Specifically, the corruption process in our method is non-Markovian, and noise is applied independently at each step rather than accumulating over time as in standard diffusion models.
> >
> > While we acknowledge the similarities that you have mentioned, we believe this distinction does not diminish the central contribution of our work. To the best of our knowledge, no existing diffusion model (for graph generation) achieves performance comparable to AFM with such a small number of denoising steps. We are open to further clarifying these points in the manuscript.
> >
> > >  The main difference seems to be the loss function. This weakens the motivation of using a filtration instead of a diffusion process.
> > >  [...]
> > > Why does your loss function perform better than DiGress' loss function? Also, I wonder what the performance will be like (i) when you train DiGress using your loss function and (ii) when you train AFM using DiGress' loss function
> >
> > Regarding your first question (i), we plan to run experiments to train DiGress using our loss function, but the results are unlikely to be available before the end of the discussion period. However, we would like to emphasize that the outcome of this experiment does not detract from our main contribution. As we have argued previously, AFM represents a distinct method rather than a variant of DiGress. Thus, findings related to DiGress' performance are orthogonal to the scope of our work.
> >
> > For your second question (ii), training AFM with the DiGress loss is not straightforward due to the differences in the underlying "noise process". In AFM, this process takes a significantly more complex form than in DiGress. Specifically, predicting $p_\theta(G_T | G_{t-1}, \dots, G_0)$ directly would require generating a filtration sequence from a sample $\tilde G_T \sim p_\theta(G_T | G_{t-1}, \dots, G_0)$. This, in turn, would require an eigendecomposition of the Laplacian of the line graph of $\tilde G_T$ in every generation step, resulting in substantial computational overhead. We do not anticipate that this approach would yield favorable results. This stands in contrast to DiGress, where the distribution $\tilde p_\theta(G_T | G_{t-1})$ can be easily pushed forward to a tractable one $p_\theta(G_t | G_{t-1})$. We hope this further clarifies distinctions between our method and DiGress.
> >
> >
> > > Table 9 in [1] shows that their continuous-time model still achieves good performance even when using only 5 steps. How does AFM compare with these continuous-time models in terms of the tradeoff?
> >
> > Table 9 of [1] (and the analagous table 6) presents performance results on the small QM9 dataset, consisting of graphs with *at most 9 nodes*. Unfortunately, the authors do not give information on the concrete hyper-parameter details for planar or SBM graphs. They only mention that the number of denoising steps was chosen from $\{50, 100, 500\}$ across all datasets. As a result, it is not possible to deduce the exact efficiency-quality tradeoff for these datasets from the information provided in [1].
> >
> > However, since DisCo-GT (the only variant that is competitive with AFM on the planar dataset) uses the same backbone architecture as DiGress, we are reasonably confident that DisCo-GT is slower during inference than AFM on the planar graph dataset, regardless of their hyperparameter choice on this dataset (based on our experiment in Figure 2).
> >
> > At this time, we note that *no code has been released for DisCo or Cometh*, which limits our ability to perform additional experiments or direct comparisons.

---

> > ### Author Response · Authors · 2024-11-28
> > **Author Response to Follow-Up Comments (2/2)**
> >
> > > my point is that the adversarial finetuning might contribute more than the proposed filtration framework. If so, this would substantially weaken the motivation of this paper. Thus, to rule out this possibility, a better ablation study is to apply adversarial finetuning to stronger baselines like DiGress.
> >
> > As shown in Table 21, a good efficiency-quality tradeoff after training stage I is crucial for satisfactory performance in stage II. Our findings suggest that other baselines are unlikely to offer a tradeoff comparable to AFM after stage I. Specifically:
> >
> > - **For GRAN**: Please refer to our previous response for details.
> > - **For DiGress**: Without adversarial finetuning, AFM achieves a VUN of 23% on the planar graph dataset using 30 steps. In contrast, as shown in Figure 2a, DiGress would require at least 45 steps to achieve a VUN > 20%. Furthermore, Figure 2b indicates that in this regime, DiGress would be significantly slower during inference than AFM. Hence, even without adversarial finetuning, AFM demonstrates a superior quality-efficiency tradeoff.
> >
> > We agree that studying the application of adversarial finetuning to existing methods is an interesting direction for future research. We would be happy to update the manuscript to mention this as a possible avenue for future investigation. However, we believe this does not diminish the importance of developing models that inherently achieve a strong quality-efficiency tradeoff without relying on adversarial finetuning. AFM is designed to address this need.

---

> > > ### Comment · Reviewer_vHcj · 2024-11-28
> > >
> > > Thank you for your detailed response.
> > >
> > > I still do not agree with the motivation of this work because "noise-augmented filtrations" are literally not filtrations. It is more like an autoregressive variant of diffusion models with the absorbing stationary distribution.
> > >
> > > Nevertheless, the authors have addressed all my other concerns. I would like to increase my score if the authors revise their terminologies to avoid confusing readers who have background in filtrations.

---

> > > > ### Author Response · Authors · 2024-11-29
> > > > **Steps to Improve Terminology**
> > > >
> > > > Thank you for your swift response!
> > > >
> > > > While we can no longer upload new revisions of our manuscript, we would be happy to incorporate your suggestions into our camera-ready version. We would like to take the following concrete steps to improve the clarity of our terminology:
> > > > - Rename our method to "Autoregressive Noisy Filtration Modeling" (or something comparable conveying the deviation from the original filtration).
> > > > - Introduce the concept of noisy filtrations earlier: Currently, we present "noise augmentation" as a part of our training algorithm. We are happy to introduce this concept earlier, namely at the end of Sec. 3.1 ("Graph Filtration"). Concretely, we would mention here that graph filtrations form the basis on which we construct stochastic graph sequences for stage I training. Details regarding this construction would remain in Sec. 3.3.1.
> > > > - Add a (possibly compressed) comparison of absorbing state diffusion models and A(N)FM to the related work section (currently we compare the two approaches in the extended related work section of the appendix).
> > > >
> > > >
> > > > If you have any further suggestions, we are happy to discuss them!

---

> > > > > ### Comment · Reviewer_vHcj · 2024-11-29
> > > > >
> > > > > Thank you for your response. I have raised my score.

---

> > > > > > ### Author Response · Authors · 2024-11-29
> > > > > >
> > > > > > Thank you for your valuable feedback and support! If you have any further questions or comments, we remain available for discussion.

---

### Official Review · Reviewer_GgH1 · 2024-11-10

**Soundness:** 2
**Presentation:** 3
**Contribution:** 2
**Rating:** 5
**Confidence:** 4

**Summary:**

This paper introduces an efficient autoregressive graph generation algorithm that leverages filtration to decompose graphs into ordered sequences of nested subgraphs each consisting of a subset of edges from the original graph. The paper explores different filtration functions and a filtration schedule sequence. The generation process first inverses the filtration process and learns the parameters of structural mixing and temporal mixing networks in a teacher-forcing fashion. Then, in the second stage, adversarial fine-tuning with reinforcement learning is employed to mitigate exposure bias of the AR process. Additionally, it uses noise augmentation to further mitigate exposure bias by randomly perturbing edges of intermediate graphs during the first stage of training. This results in speedup in the graph sampling from the proposed AR model.

**Strengths:**

- The paper proposes a novel autoregressive graph generation based on filtration and employs various techniques to improve generation performance.
- The graph generation idea is sound and appealing.
- The paper is well-written and provides clear justification for the core elements of the proposed graph generative process, including using adversarial fine-tuning, noise augmentation, filtration functions and filtration scheduling.

**Weaknesses:**

1. **Lack of Complexity Analysis**: While the paper mentions that the drawback of the available diffusion and AR models is that “their quadratic or higher computational complexity with respect to the number of nodes” (which is not exact), it does not provide complexity analysis which is crucial for such work. A section or table comparing the training and sampling complexity with baseline models is recommended.
    - Regarding non-monotonic graph sequences, the complexity will increase quadratically if the algorithm needs to decide on all the possible edges rather than just adding new subsets to the existing ones.
2. While the main part of the paper is discussing the filtration process, but the ablation shows that many tricks, including noisy edge augmentation, stochastic perturbations to node orderings, and Adversarial Fine-tuning with RL, are crucial to the overall graph sampling performance and AR graph generation based on filtering, without these additions performs worse than other AR-based baselines such as GRAN. So, the effect of such techniques on other AR-based models is recommended to be investigated for a fair comparison.
3. **Missing Citation and Comparison**: The paper lacks citations and comparisons to several relevant graph generation models. For example, HiGen also offers a scalable hierarchical graph generation in autoregressive manner in a sequence of coarse to fine subgraphs which captures multi-scale properties of the graph. Also other graph diffusion models such as GDSS [2] and EDGE [3] are not cited.
4. **Insufficient Empirical Studies**: The experiments presented are not sufficient to fully demonstrate the capabilities of the proposed model. In many cases, the performance of AFM falls short compared to diffusion models and sometimes even AR counterparts.
   - a. While this paper claims to improve efficiency, it lacks experiments for large graphs (graphs with >1k ). Evaluating the performance on larger graphs is crucial to assess its scalability.
   - b. The method can provide a trade-off between expressiveness and efficiency. Investigating how a larger sequence of subgraphs affects performance and determining the optimal sequence length particularly for large graphs would be valuable.
5. The choice of filtration function may affect the scheduling method. Also, An analysis of different scheduling methods with various filtration functions is recommended to understand their interplay.
6. One potential limitation of this approach is its edge independence [4]. The proposed method's approach of adding many edges simultaneously could lead to the edge-independence issue.
7. More clarification is needed regarding the adversarial fine-tuning with RL. The reward used in PPO should be specified, and the figure illustrating the process needs clarification. Additionally, a more detailed explanation of the "TRAINGENERATORANDVALUEMODEL" pseudo-code is necessary for better understanding.

**Questions:**

1. The proposed method learns the joint likelihood of the sequence (equation 4). However, graph generative models aim to learn $P_{\theta}(G_T)$. The paper should clarify the relationship between these two, especially considering the use of noise augmentation and multiple trajectories in the training set.
2. What are the training complexity and sampling complexity of the proposed model?
3. AR models' sensitivity to node ordering is a known drawback, is the proposed AFM order equivariant? While the paper mentions that the proposed AFM depends on the choice of edge ordering, this point requires additional clarification and empirical experimentation.
4. Since the line Fiedler function outperforms the other two options and is preferred in your experiments, then does it infer that structural consistency is prioritized? How does it affect the graph generation?
5. In Page 1, line 48 needs a citation.



Conclusion:

While the proposed idea is very appealing, and I acknowledge its potential impact, I am not sure that the paper is ready for ICLR and as impactful as it could be in its current form. Therefore, I am hesitant to support acceptance but I am willing to reconsider my score if my concerns and questions are addressed.


References:
[1] M. Karami, “HiGen: Hierarchical Graph Generative Networks”, International Conference on Learning Representations, 2024.
[2] Jaehyeong Jo, Seul Lee, and Sung Ju Hwang. Score-based generative modeling of graphs via the system of stochastic differential equations. In International Conference on Machine Learning, pp. 10362–10383. PMLR, 2022.
[3] Xiaohui Chen, Jiaxing He, Xu Han, and Li-Ping Liu. Efficient and degree-guided graph generation via discrete diffusion modeling. arXiv preprint arXiv:2305.04111, 2023.
[4] Sudhanshu Chanpuriya, Cameron Musco, Konstantinos Sotiropoulos, and Charalampos Tsourakakis. On the power of edge independent graph models. Advances in Neural Information Processing Systems, 34: 24418–24429, 2021.

---

> ### Author Response · Authors · 2024-11-24
> **Author Response to Reviewer GgH1 (1/3)**
>
> Thank you for your time and effort in providing such constructive feedback. We greatly appreciate your insights and have addressed the points you raised below.
>
> ## Weaknesses
> > Lack of Complexity Analysis
>
> We have addressed this point in the "Complexity Analysis" section in our general response.
>
> > the complexity will increase quadratically if the algorithm needs to decide on all the possible edges
>
> The computational complexity of AFM increases even cubically due to the computation of Laplacian positional embeddings. The DiGress model exhibits the same complexity w.r.t. $N$. While our asymptotic runtime complexity appears less favorable compared to other autoregressive models (e.g. GraphRNN, GRAN), our empirical findings challenge this theoretical limitation. Notably, on protein graphs with up to 500 nodes, our approach demonstrated substantially faster inference times than baseline models.
> Moreover, the efficient performance of AFM remains consistent across different dataset scales (they are all around 0.03s/graph), suggesting that its efficiency is primarily driven by its ability to use a small $T$.
>
>
> > Missing Citation and Comparison
>
> In our revised version we discuss, among others, HiGen, GDSS, and EDGE in the related work section. We draw more detailed comparisons in the extended related work section of the appendix. We have also added results for HiGen and EDGE to Table 1. Our main conclusions remain unchanged.
>
> >  In many cases, the performance of AFM falls short compared to diffusion models and sometimes even AR counterparts.
>
> We respectfully disagree with the assertion that AFM falls short compared to AR counterparts. Our experimental results indicate that AFM generally outperforms existing AR methods. Below, we summarize our comparisons with other AR models:
>
> - **Protein graphs**: AFM outperforms the state-of-the-art AR baseline (GRAN) w.r.t. all MMD metrics.
> - **Expanded synthetic datasets**: Across all expanded synthetic datasets, AFM outperforms GRAN w.r.t. VUN and all MMD metrics.
> - **Small synthetic datasets**: AFM outperforms both GraphRNN and GRAN in terms of the VUN metric. Regarding the MMD metrics, no AR method consistently dominates across all graph descriptors. It's important to note that MMD estimates on these small datasets have high variance due to the limited sample size (40 samples). As we demonstrate in Appendix "A.17.2 Bias and Variance of MMD Estimation", the standard deviation of the MMD estimates can be even comparable to the estimates themselves. This high variance was a primary motivation for studying the expanded synthetic datasets, where MMD metrics exhibit substantially lower variance and provide more reliable comparisons.
>
>
> > While this paper claims to improve efficiency, it lacks experiments for large graphs
>
> We would like to clarify that some of the protein graphs used in our experiments are already moderately large, containing up to 500 nodes.
> Currently, we are not aware of publicly available datasets with a reasonable sample size that would allow us to properly benchmark our method on graphs consisting of more than 1K nodes.
>
> - Point Cloud dataset by Liao et al. (2020): This dataset includes graphs with up to 5,000 nodes but contains only 26 graphs in the training set and 8 graphs in the test set. Training a generative model and performing a thorough evaluation on such a small dataset is challenging due to high variance and limited statistical significance.
> - Single Large Graphs (e.g., EDGE with CiteSeer or Cora): Some works have trained models on individual large graphs. However, this approach focuses on generating a single graph rather than learning a distribution over graphs, which is the primary objective of our work.
> - Subgraphs from Larger Networks (e.g., GraphARM): Other approaches sample subgraphs from larger citation networks. Unfortunately, these subgraphs are generally small, with sizes up to 87 nodes in the case of the Cora dataset used by GraphARM. This does not address the need for experiments on significantly larger graphs.
>
> If there are specific datasets that you can recommend for evaluating generative models on larger graphs, we would be eager to consider them. However, we also want to clarify that our goal is to demonstrate the **good balance between efficiency and generation quality** of our method in high-throughput applications, rather than the scalability to generating larger graphs. To clarify this point, we have revised our manuscript to replace the term "scalability" with the more precise term "efficiency" and "high-throughput generation" (see lines 24, 64, 74 of original submission).
>
>
> > Investigating how a larger sequence of subgraphs affects performance and determining the optimal sequence length particularly for large graphs would be valuable.
>
> Please refer to the "Effect of the filtration granularity $T$" section in our general response.

---

> ### Author Response · Authors · 2024-11-24
> **Author Response to Reviewer GgH1 (2/3)**
>
> ## Weaknesses
>
> > The choice of filtration function may affect the scheduling method. Also, An analysis of different scheduling methods with various filtration functions is recommended to understand their interplay.
>
> The extensive landscape of potential filtration and scheduling function combinations, coupled with their potential sensitivity to dataset and other hyperparameter selections, prevents definitive generalized recommendations. Nevertheless, we conduct separate ablation studies on the expanded planar graph dataset for both filtration and scheduling functions in Appendix A.16, providing some insights into their individual contributions when fixing the other hyperparamter. We find that (in combination with the linear schedule), the line Fiedler filtration function appears to perform best. In combination with the line Fiedler function, the concave schedule appears to perform best w.r.t. the VUN metric, followed by the linear schedule.
>
>
> > One potential limitation of this approach is its edge independence [4]. The proposed method's approach of adding many edges simultaneously could lead to the edge-independence issue.
>
> Unfortunately, the reference [4] appears to be missing in your review. We suppose that you refer to the fact that the distribution $p_\theta(E_t | G_{t - 1}, \dots, G_{0})$ should not be decomposed as a product over all edges (i.e. the edge presence should not be independent). We agree with this statement, as edges may, for instance, exhibit mutual exclusivity constraints (e.g. in tree graphs that do not contain cycles). For this reason, we note that we approximate the distribution above as a mixture of multivariate Bernoulli distributions, thereby introducing dependencies.
>
> > More clarification is needed regarding the adversarial fine-tuning with RL
>
> Following your suggestions, we have extended the section in the appendix to provide details on the adversarial fine-tuning (please see "A.18 Adversarial Finetuning Details"). We have added a reference to this appendix in Figure 1 of the main text. If you have any concrete suggestions on improving clarity further, we are happy to discuss and incorporate them.

---

> ### Author Response · Authors · 2024-11-24
> **Author Response to Reviewer GgH1 (3/3)**
>
> ## Questions
> > The proposed method learns the joint likelihood of the sequence. However, graph generative models aim to learn $p_\theta(G_T)$. The paper should clarify the relationship between these two
>
> We have addressed this concern in the appendix of the revised paper for training stage I (see "A.4 A Bound on Model Evidence"). We show that the autoregressive loss in stage I serves as an evidence lower bound for the marginal $p_\theta(G_T)$. For training stage II, we note that we no longer consider the joint likelihood of the sequence but only present the final samples $G_T$ to the discriminator. It is known that the resulting training criterion is the Jensen-Shannon divergence between $p_\theta(G_T)$ and the data distribution (see, e.g., [this paper](https://proceedings.neurips.cc/paper_files/paper/2014/file/5ca3e9b122f61f8f06494c97b1afccf3-Paper.pdf)).
>
> > What are the training complexity and sampling complexity of the proposed model?
>
> Please refere to the "Complexity Analysis" section in our general response regarding the sampling complexity.
>
> Regarding training complexity: In AFM, a forward and backward pass on a single graph sample is $\mathcal{O}(T^2N + TN^3)$ in stage II and $\mathcal{O}(T^2N +TN^2)$ in stage I, assuming that Laplacian positional encodings are pre-computed for stage I. However, we want to clarify that these complexities give no indication of how quickly the model can be trained, as the rate of convergence during training may be governed by various other factors.
>
>
> > AR models' sensitivity to node ordering is a known drawback, is the proposed AFM order equivariant?
>
> Our proposed Mixer architecture is permutation equivariant. I.e., if $\pi \in S_N$ is a permutation of nodes, we have $p_\theta(\pi(G_t) | \pi(G_{t-1}), \dots, \pi(G_0)) = p_\theta(G_t | G_{t-1}, \dots, G_0)$.
> However, we do enrich the input node representations of the empty graph $G_0$ with positional embeddings derived from a pre-defined node ordering. During training, this ordering is extracted from the input graph $G_T$, while inference allows for arbitrary ordering due to the model's equivariance property. A detailed exploration of these input node representations can be found in the appendix (please see "A.6 Input Node Representations" for more details). We investigate the effect of this node ordering in our ablation studies ("A.16 Additional Ablations"). While, generally, we find it to improve performance after stage I training, it is not a critical component in all settings. We remain open to conducting additional stage II training with these ablation variants if you consider it essential.
>
>
> > While the paper mentions that the proposed AFM depends on the choice of edge ordering, this point requires additional clarification and empirical experimentation.
>
> With "edge ordering", we refer to the order in which edges are deleted in the filtration process. Hence, this ordering is determined entirely by the filtration function. We present experiments on different choices in the appendix ("Additional Ablations").
>
> "Edge ordering" is mentioned in our original manuscript in line 93 of the related work section. We attempt to improve clarity by explicitly stating that the edge ordering is induced by the choice of filtration function.
>
> If you have concrete suggestions for further experiments on this question, we are open to conducting them.
>
> > Since the line Fiedler function outperforms the other two options and is preferred in your experiments, then does it infer that structural consistency is prioritized? How does it affect the graph generation?
>
> Ultimately, the line Fiedler filtration function is a heuristic choice and we don't have a rigorous explanation as to why it works well. Intuitively, the line Fiedler function paints edges of a graph in a "continuous gradient", assigning similar values to neighboring edges. We hypothesize that this may simplify the task of learning to reverse the filtration, as edges are mostly added in locally confined regions that neighbor the edges that were added in the previous steps.
> To guide your intuition, we refer to Figure 5 ("Visualization of different filtration functions on a planar graph") for an illustration of the behavior of the line Fiedler function.
>
>
> > In Page 1, line 48 needs a citation.
>
> We have added citations to Edelsbrunner et al. and Zomordian & Carlsson.

---

> ### Comment · Reviewer_GgH1 · 2024-12-03
>
> I would like to thank the authors for their detailed response and for providing an updated version which helps in addressing most of the concerns. However, I still have some concerns that still need consideration:
>
> - Given the existence of other AR-based methods for graph generations, such as HiGen, with  comparable to (or even better that) some diffusion models, I discourage using the repeated claim that this work is the *"first AR model that matches the performance of diffusion-based models while maintaining its speed advantage"*l.
>
> - It is recommended to include the performance and sampling time of HiGen [1] on the Protein dataset, noting that the sampling speedup was reported in the appendix.
>
> - As raised by another reviewer, the adversarial training stage (stage II) can be applied to other generative models for performance improvement, as it is not an integral part of this architecture. Therefore, a fairer comparison would involve comparing the proposed model with other models augmented with stage II, such as Digress+Stage II, Gran+stage II, HiGen+stage II, and others.
>
> - To evaluate the sensitivity of the proposed model to node-ordering, it would be beneficial to observe its performance using canonical node orderings like BFS, DFS and so on.
>
> Please also note that my initial review is updated to include the missing references.
>
> References:
> [1] M. Karami, “HiGen: Hierarchical Graph Generative Networks”, International Conference on Learning Representations, 2024.

---

> > ### Author Response · Authors · 2024-12-03
> > **Author Response to Further Comments (1/2)**
> >
> > Thank you for considering our rebuttal and manuscript revisions. We provide below our responses to your additional comments:
> >
> > > Given the existence of other AR-based methods for graph generations, such as HiGen, with comparable to (or even better that) some diffusion models, I discourage using the repeated claim that this work is the *"first AR model that matches the performance of diffusion-based models while maintaining its speed advantage"*.
> >
> > After further consideration, we recognize the importance of a closer comparison with HiGen, given its competitive performance and efficiency improvements. We agree that revising our claim is appropriate in light of HiGen's results. Based on this consideration, we propose the following revised statement:
> >
> > "To the best of our knowledge, we present the first AR model that matches the performance of diffusion-based models (particularly in terms of VUN metrics) on synthetic Planar and SBM datasets while maintaining its inference speed advantage."
> >
> > Our rationale for this claim is as follows: we are indeed unaware of other autoregressive models that report competitive performance on these datasets. HiGen does not report results on the planar graph dataset or the VUN metric on the SBM dataset. These datasets are, in our view, among the most widely studied and relevant benchmarks in the field of graph generation, and have played a critical role in establishing DiGress as a robust baseline. Furthermore, the hierarchical nature of HiGen raises questions about its generalizability to non-hierarchical graph structures like planar graphs. As such, while we acknowledge HiGen's strong performance in specific scenarios, its applicability to the more widely used datasets and metrics we emphasize remains unclear.
> >
> >
> > > It is recommended to include the performance and sampling time of HiGen [1] on the Protein dataset, noting that the sampling speedup was reported in the appendix.
> >
> >
> > A direct comparison between HiGen and AFM is unfortunately not feasible without re-training HiGen, as the inference speed of HiGen was measured on different hardware. Additionally, we compute our MMD metrics on 1024 generated graphs, due to the high variance we observed with fewer graphs (see Appendix A.17) while HiGen only used 40 and 183 generated graphs on the SBM and protein datasets, respectively. While we are actively preparing a comparison to HiGen, these experiments will not be completed before the discussion period ends.
> >
> > While a rigorous comparison between AFM and HiGen is not possible at this time, we include a table below showing the inference times reported by HiGen alongside those of AFM on shared datasets. We also compare the reported speedup relative to GRAN.
> >
> >
> > | Dataset  | Model       | Sampling Time* ($\downarrow$) | Speedup over GRAN ($\uparrow$) |
> > |----------|-------------|-----------------------------|--------------------------------|
> > | Protein  | HiGen       | 1.33                       | 34.6x                         |
> > |          | AFM         | 0.194                      | 11.6x                         |
> > | SBM      | HiGen       | 0.4653                     | 3.4x                          |
> > |          | AFM         | 0.0301                     | 4.4x                          |
> >
> > \* No direct comparisons possible
> >
> >
> > We find that HiGen reports higher inference times for both HiGen and GRAN compared to our results. However, the sources of this discrepancy is not clear. It may stem from differences in hardware (HiGen appears to perform inference on CPU while AFM uses GPU), inference settings (such as batch size etc.) or interpretation of sampling time (we normalize to a per-graph cost, whereas HiGen's time measurement approach is not specified).
> >
> > Regarding the speedup over GRAN, we find that HiGen achieves a larger improvement on the protein dataset while AFM achieves a larger improvement on the SBM dataset.

---

> > ### Author Response · Authors · 2024-12-03
> > **Author Response to Further Comments (2/2)**
> >
> > > As raised by another reviewer, the adversarial training stage (stage II) can be applied to other generative models for performance improvement
> >
> > As shown in Table 21, a good efficiency-quality tradeoff after training stage I is crucial for satisfactory performance in stage II. Our findings suggest that other baselines are unlikely to offer a tradeoff comparable to AFM after stage I. Specifically:
> >
> > - **For GRAN**: In all of our experiments, AFM is faster during inference than GRAN. In particular on larger protein graphs, AFM is ten times faster. This is not affected by adversarial finetuning. After training stage I, we already observe that AFM outperforms GRAN substantially in terms of quality (VUN) on the expanded planar graph dataset (c.f. Table 4, "Stage I w/ Noise"). The same holds true for the expanded SBM dataset (c.f. Table 15, "A.16 Additional Ablations"). AFM without adversarial finetuning performs slightly worse than GRAN on the expanded lobster dataset (c.f. Table 16).
> > - **For DiGress**: Without adversarial finetuning, AFM achieves a VUN of 23% on the planar graph dataset using 30 steps. In contrast, as shown in Figure 2a, DiGress would require at least 45 steps to achieve a VUN > 20%. Furthermore, Figure 2b indicates that in this regime, DiGress would be significantly slower during inference than AFM. Hence, even without adversarial finetuning, AFM demonstrates a superior quality-efficiency tradeoff.
> >
> >
> > We agree that studying the application of adversarial finetuning to existing methods is an interesting direction for future research. We would be happy to update the manuscript to mention this as a possible avenue for future investigation. However, we believe this does not diminish the importance of developing models that inherently achieve a strong quality-efficiency tradeoff without relying on adversarial finetuning. AFM is designed to address this need.
> >
> > > To evaluate the sensitivity of the proposed model to node-ordering, it would be beneficial to observe its performance using canonical node orderings like BFS, DFS and so on.
> >
> > We have launched an experiment to evaluate the performance under a DFS ordering. However, we note that our intention behind the introduced node ordering was to provide prior information on the filtration we chose. Hence, we would not expect an ordering chosen independently of the filtration function to be beneficial. We will provide an update if the results of this experiments become available before the end of the discussion period.
> >
> > We also refer to Tables 19 where we study various different orderings. From these experiments we found that introducing an ordering is not always necessary to achieve satisfactory performance

---

> > ### Author Response · Authors · 2024-12-04
> > **Experiments for DFS Node Ordering**
> >
> > Below, we report the results for stage I training with a DFS ordering on the expanded planar graph dataset (after 100k steps). This is an extension of table 19.
> >
> > | Node embedding      | VUN (↑)    | Degree (↓) | Clustering (↓) | Spectral (↓) | Orbit (↓)  |
> > | ---------------- | ---------- | ---------- | -------------- | ------------ | ---------- |
> > | DFS ordering      | **28.91%** | **0.0055** | 0.1802         | 0.0024       | **0.0053** |
> > | Current ordering in the paper | 20.21%     | 0.0058     | **0.1768**     | 0.0048       | 0.0129     |
> > | Random ordering  | 18.36%     | 0.0085     | 0.2332         | **0.0023**   | 0.0091     |
> > | Gaussian noise   | 12.99%     | 0.0086     | 0.2356         | 0.0031       | 0.0112     |
> > | No embedding      | 13.48%     | 0.0057     | 0.2195         | 0.0023       | 0.0091     |
> >
> > Judging by VUN, we find that the DFS variant appears to perform even better than the other variants we studied previously. We are happy to perform more extensive ablation studies with other ordering schemes for the camera-ready version. We thank the reviewer for this interesting suggestion.
> >
> > However, we want to clarify again that the *node ordering is a separate component from the filtration function*. The filtration function determines the order in which edges are added while the node ordering provides prior information on how nodes should be positioned in the sampled graph $G_T$ (we refer to A.6 for details on how we transform the ordering into node-wise embeddings). As becomes apparent from the "random ordering" and "no embedding" experiments above, this prior information is not always necessary to obtain satisfactory performance. In this sense, the node ordering plays a different role in our work than in GRAN or GraphRNN.

---

### Official Review · Reviewer_7YaU · 2024-11-10

**Soundness:** 1
**Presentation:** 2
**Contribution:** 2
**Rating:** 3
**Confidence:** 4

**Summary:**

This paper proposes an autoregressive based graph generation by sequentially adding edges until the graph is completed. The author uses "graph filtration" to define the autoregressive edge sequence, modeling edge dependency with multivariant bernoulli distribution, adding noise to each next-block prediction task to reduce the exposure bias of autoregressive method, and finally using GAN based adversarial training to further improve the generation quality. Experiments show the speed improvement comparing with diffusion based graph generation, but the performance is not improved much. The author also discussed the limitations of the proposed method, such as not supporting general attributed graphs with node labels and edge labels.

**Strengths:**

1. The direction of autoregressive graph generation is important given its superior speed to diffusion models.
2. The paper contains extensive designs for many issues raised in autoregressive graph generation, such as designing the order, exposure bias, multivariable dependency, and the generator issue caused during iterative generation.
3. The author discussed their method's limitation in detail, which helps to clarify several important questions.

**Weaknesses:**

1. Some part of the method is not clear or containing error:

* eq. 7 left side is conditional on t+1 to T steps, which is not correct.
* while figure 1 shows the illustration of training procedures, the generation procedure is not mentioned, as well as the detailed setting of the generator during GAN training.

2. Some part of design is not efficient:
* the mixer architecture design uses too many representations: T x N with N being the total number of nodes. This can cause significantly increased memory requirement.
* many designs are not general, such as not supporting attributed graphs with node labels and edge labels, which are well supported by diffusion models and traditional AR graph generation methods.

3.  The designed method is a bit complicated: with many designs in autoregressive part, and additional adversarial training for further performance improvement. However, when comparing to baselines, I cannot find consistent and noticeable performance improvement, although the method is faster than diffusions.

4. The author miss some important and highly related references and baselines, such as the [1]. They share many common parts and should be discussed.

5. Limited experiments: only simple and small simulation graphs.

[1] Zhao, Lingxiao, Xueying Ding, and Leman Akoglu. "Pard: Permutation-Invariant Autoregressive Diffusion for Graph Generation." arXiv preprint arXiv:2402.03687 (2024).

**Questions:**

Except these weakness, can you provide some possibilities of extending the current method to attributed graphs?
If it cannot show its application to graphs like molecules, the usage and scope of the proposed method is very limited.

---

> ### Author Response · Authors · 2024-11-24
> **Author Response to Reviewer 7YaU (1/2)**
>
> Thank you for your time and effort in providing constructive feedback. We appreciate your insights and have addressed the points you raised below.
>
> ## Weaknesses
> > eq. 7 left side is conditional on t+1 to T steps, which is not correct.
>
> We thank the reviewer for pointing out this typo. It is corrected in our revised manuscript.
>
> > while figure 1 shows the illustration of training procedures, the generation procedure is not mentioned, as well as the detailed setting of the generator during GAN training.
>
> The generation procedure follows the standard autoregressive approach: Namely, we start with the empty graph $G_0$ and iteratively sample $G_t$ from the learned distribution $p_\theta(G_t | G_{t - 1}, \dots, G_0)$.
>
> We have updated the figure caption with a reference to the appendix in which we provide additional in-depth description of the GAN training, including the inference setting of the generator.
>
> > the mixer architecture design uses too many representations: T x N with N being the total number of nodes. This can cause significantly increased memory requirement.
>
> We acknowledge the concern that the mixer architecture may require significant memory if $T$ is large. However, in our experiments, we have not found memory complexity to be an issue for the graph sizes we considered, due to our ability to use a small $T$. To address potential memory constraints, we have studied a simplified autoregressive structure with a memory complexity of $\mathcal{O}(N)$, assuming efficient implementations of attention mechanisms. This alternative is detailed in Appendix "A.3 A First-Order Autoregressive Variant." Our additional experiments demonstrate that this simplified structure is a viable option if memory consumption becomes a concern in practical applications. Please also refer to the "Complexity analysis" section in our response for a more detailed discussion.
>
> > many designs are not general, such as not supporting attributed graphs with node labels and edge labels, which are well supported by diffusion models and traditional AR graph generation methods.
>
> We acknowledge the concern that our current design does not explicitly support graphs with node and edge attributes. However, we want to emphasize that our primary focus is on developing efficient AR model for unattributed graphs and we present the **first AR model that matches the performance of diffusion-based models while maintaining its speed advantage**. Furthermore, un-attributed graph topologies are often of significant interest in various applications, such as protein structure modeling and design, where contact maps represent the spatial proximity between amino acids [1, 2].
>
>
> As discussed in our conclusion and appendix, our method is naturally extendable to edge-labeled graphs. Regarding the modeling of node attributes, we propose two prospective approaches:
> - Post-hoc node labeling: After generating the graph topology, node labels can be assigned based on the structural context, as suggested by Bergmeister et al. (2024). For example, in molecular graphs, valency constraints and local connectivity patterns may provide strong cues for determining atom types. Such an approach was used in [3].
> - Extension of the Filtration Process: We can extend the filtration process to include node labels by representing them as labeled self-loops and incorporating them into the edge filtration. This approach allows the model to learn node attributes alongside the graph topology within the same generative framework.
>
> We believe that integrating node and edge labels is a promising direction for future work, which can further enhance the applicability of our approach to a broader range of graph generation tasks.

---

> ### Author Response · Authors · 2024-11-24
> **Author Response to Reviewer 7YaU (2/2)**
>
> ## Weaknesses
>
> > The designed method is a bit complicated: with many designs in autoregressive part, and additional adversarial training for further performance improvement.
>
> We acknowledge that our proposed method involves multiple components and may appear technically complex. However, our ablation studies demonstrate that each of the introduced techniques is essential for achieving the observed performance improvements. We believe that these technical contributions not only enhance our model but could also **have applications in other generative models within the field**.
>
> To facilitate adoption and ease of use, we have added a section to the appendix giving practical advice on the choice of hyperparameters ("A.5 Practical Advice on Hyperparameter Choice"). Additionally, we are committed to transparency and reproducibility; therefore, we will publish our source code upon acceptance of the paper. We hope that these resources will help the community in implementing and extending our work.
>
> > However, when comparing to baselines, I cannot find consistent and noticeable performance improvement, although the method is faster than diffusions.
>
> We present an autoregressive model that achieves state-of-the-art performance on several benchmarks while being orders of magnitude faster than competing diffusion models. We do not aim to outperform these diffusion baselines in terms of quality but focus our efforts on matching their quality while substantially improving inference speed.
> **To the best of our knowledge, our method is the first autoregressive graph generative model that can compete with diffusion models on the benchmarks we presented.**
>
> > The author miss some important and highly related references and baselines, such as the [1]. They share many common parts and should be discussed.
>
> Thank you for this suggestion. We discuss the similarities and differences of AFM and Pard in the extended related work section of the appendix ("A.1 Extended Related Work").
>
> > Limited experiments: only simple and small simulation graphs.
>
> We would like to clarify that we presented our method on real-world protein graphs that contain up to 500 nodes.
> Currently, we are not aware of publicly available datasets with a reasonable sample size that would allow us to properly benchmark our method on graphs consisting of more than 1K nodes.
>
> - Point Cloud dataset by Liao et al. (2020): This dataset includes graphs with up to 5,000 nodes but contains only 26 graphs in the training set and 8 graphs in the test set. Training a generative model and performing a thorough evaluation on such a small dataset is challenging due to high variance and limited statistical significance.
> - Single Large Graphs (e.g., EDGE with CiteSeer or Cora): Some works have trained models on individual large graphs. However, this approach focuses on generating a single graph rather than learning a distribution over graphs, which is the primary objective of our work.
> - Subgraphs from Larger Networks (e.g., GraphARM): Other approaches sample subgraphs from larger citation networks. Unfortunately, these subgraphs are generally small, with sizes up to 87 nodes in the case of the Cora dataset used by GraphARM. This does not address the need for experiments on significantly larger graphs.
>
> If there are specific datasets that you can recommend for evaluating generative models on larger graphs, we would be eager to consider them. However, we also want to clarify that our goal is to demonstrate the **good balance between efficiency and generation quality** of our method in high-throughput applications, rather than the scalability to generating larger graphs. To clarify this point, we have revised our manuscript to replace the term "scalability" with the more precise term "efficiency" and "high-throughput generation" (see lines 24, 64, 74 of original submission).
>
>
> ## Questions
>
> > can you provide some possibilities of extending the current method to attributed graphs
>
> We have provided propspective avenues for future research above.
>
> ## References
> [1] Zeming Lin et al., Evolutionary-scale prediction of atomic-level protein structure with a language model. Science379, 1123-1130(2023). DOI:10.1126/science.ade2574
>
> [2] Chen, Y., Xu, Y., Liu, D. et al. An end-to-end framework for the prediction of protein structure and fitness from single sequence. Nat Commun 15, 7400 (2024). https://doi.org/10.1038/s41467-024-51776-x
>
> [3] Li, Yibo, Jianxing Hu, Yanxing Wang, Jielong Zhou, Liangren Zhang, and Zhenming Liu. ‘DeepScaffold: A Comprehensive Tool for Scaffold-Based De Novo Drug Discovery Using Deep Learning’. Journal of Chemical Information and Modeling 60, no. 1 (2020): 77–91. https://doi.org/10.1021/acs.jcim.9b00727.

---

> > ### Comment · Reviewer_7YaU · 2024-12-01
> > **Response**
> >
> > I thank the author for providing the complexity analysis and markov-independent version of the method. Nevertheless, the problems left in the paper still need a major revision.
> >
> > 1. The extension to labeled and attributed graph is important, given that the main application of graph generation is in molecular domain.
> >
> > 2. The reference and discussion to Pard [1] should be revised: first, Pard is NOT a concurrent work. Pard is available at the beginning of Feb 2024, which is 7-8 months before the ICLR deadline. The reference to Pard should appear in the main context, and the way of decompose graphs into sequences used by Pard, should be properly referenced given its similarity to the graph filtration process used in this paper.
> > - Also, the author should not claim that the proposed method is "first autoregressive graph generative model that can compete with diffusion models", given that Pard outperforms diffusion models significantly.
> > - It would also be good to include that as a baseline to compare, given their design overlap.
> >
> > 3. The performance gap between Stage I and Stage II is large for many datasets shown in the paper. This actually raises a big concern for the designed procedure: most of the paper is introducing the stage I's design while only half of page is discussing the adversarial training. However, it seems that (1) the adversarial training is not tight with the stage I's method, and can be used for other generative model as well, which indicates the problem of why combining adversarial training with the proposed method, (2) the adversarial training actually boosts the performance significantly, indicates that the design in stage I is not effective. The author needs to revise the paper to clarify the design and where the performance improvement coming from. If the adversarial training is the killer, the author should study how it applied to other methods like DiGress and Pard, given that they are solely stage I method without adversarial training. The combination of adversarial training with the primary Stage I algorithm feels like an engineering trick to bolster paper writing, especially since the core Stage I method on its own appears insufficiently robust.
> >
> >
> > [1] Zhao, Lingxiao, Xueying Ding, and Leman Akoglu. "Pard: Permutation-Invariant Autoregressive Diffusion for Graph Generation." arXiv preprint arXiv:2402.03687 (NeurIPS 2024).

---

> > > ### Author Response · Authors · 2024-12-02
> > > **Author Response to Further Comments (1/2)**
> > >
> > > Thank you for considering our rebuttal and manuscript revisions. We provide below our responses to your additional comments:
> > >
> > > > The extension to labeled and attributed graph is important, given that the main application of graph generation is in molecular domain.
> > >
> > > As noted in the 'Scope of the Work' section of our rebuttal, our current study focuses on a foundational contribution to graph generation, which we believe offers substantial value to the community. While incorporating this feature is beyond the scope of the present work, we recognize its significance and consider it a promising direction for future research.
> > >
> > > > first, Pard is NOT a concurrent work. Pard is available at the beginning of Feb 2024, which is 7-8 months before the ICLR deadline.
> > >
> > > We base our statement regarding concurrency on the official ICLR guidelines (https://iclr.cc/Conferences/2025/FAQ). According to these guidelines, works published (i.e., at a *peer-reviewed venue*) within the four months preceding the submission deadline are not considered mandatory for comparison. Nonetheless, we appreciate the reviewer’s concern and are happy to discuss the methodological differences between our work and Pard in detail below.
> > >
> > > > Also, the author should not claim that the proposed method is "first autoregressive graph generative model that can compete with diffusion models", given that Pard outperforms diffusion models significantly.
> > >
> > > We note that **Pard uses diffusion components within its architecture** and frames itself as a hybrid model combining diffusion and autoregression. As stated in its abstract: "We introduce PARD, a Permutation-invariant AutoRegressive **Diffusion model** that integrates diffusion models with autoregressive methods [...] where each block’s probability is conditionally modeled by a shared diffusion model with an equivariant network".
> > > Given this explicit hybrid nature, we believe our claim as the first purely autoregressive graph generative model that competes with diffusion models remains valid.
> > >
> > > > similarities and differences of PARD and AFM
> > >
> > > As discussed in the extended related work section, Pard does not share any substantial technical features with our work. The only apparent similarity is the fact that both Pard and our method incrementally build sequences of increasingly large subgraphs. However, this approach is not unique to Pard but is also employed by other models, e.g. GRAN. While Pard constructs a sequence of induced subgraphs based on a partial _node ordering_ (as GRAN does), our method focuses on a partial _edge ordering_.
> > >
> > > We emphasize that our approach aligns more closely with other 'edge-based' methods (e.g., Goyal et al., 2020; Bacciu et al., 2020) than with 'node-ordering' models like Pard, GRAN, GraphRNN, and GraphArm, as detailed in the related work section.
> > >
> > > To make the differences and similarities between Pard and AFM more explicit, we provide the following table:
> > >
> > > | Feature           | Pard                                                                                                                                     | AFM                                                                                                                            |
> > > | ----------------- | ---------------------------------------------------------------------------------------------------------------------------------------- | ------------------------------------------------------------------------------------------------------------------------------ |
> > > | Introduced order  | Partial order of *nodes* based on weighted sum of k-hop degrees.                                                                         | Various partial orders of *edges*. Major focus on order derived from spectral features of the line graph.                      |
> > > | Graph sequence    | Sequence of *induced* subgraphs. Monotonic. Deterministic.                                                                               | Sequence of arbitrary subgraphs. Growing but not strictly monotonic. Stochastic via noise augmentation.                        |
> > > | Sequence modeling | Likelihood-free via a diffusion model.                                                                                                   | Likelihood-based via a mixture of Bernoullis.                                                                                  |
> > > | Focus of work     | Favorable combination of diffusion and autoregression. Efficiency is mentioned but no inference time comparisons are provided. | Autoregressive model competing with diffusion in terms of quality, with inference efficiency as the main goal (supported by generation time comparisons). |
> > >
> > >
> > >
> > > > It would also be good to include that as a baseline to compare, given their design overlap.
> > >
> > > Please refer to our above response regarding "concurrency" and "methodological differences".

---

> > > ### Author Response · Authors · 2024-12-02
> > > **Author Response to Further Comments (2/2)**
> > >
> > > > the way of decompose graphs into sequences used by Pard, should be properly referenced given its similarity to the graph filtration process used in this paper.
> > >
> > > We appreciate the reviewer’s suggestion and will gladly reference Pard regarding this aspect in the extended related work section. However, we emphasize that the two approaches for constructing graph sequences are fundamentally distinct given the methodological differences discussed above.
> > >
> > >
> > > > The performance gap between Stage I and Stage II is large for many datasets shown in the paper. This actually raises a big concern for the designed procedure: most of the paper is introducing the stage I's design while only half of page is discussing the adversarial training.
> > >
> > >
> > > As shown in Table 21, a good efficiency-quality tradeoff after training stage I is crucial for satisfactory performance in stage II. Our findings suggest that other baselines are unlikely to offer a tradeoff comparable to AFM after stage I. Specifically:
> > >
> > > - **For GRAN**: In all of our experiments, AFM is faster during inference than GRAN. In particular on larger protein graphs, AFM is ten times faster. This is not affected by adversarial finetuning. After training stage I, we already observe that AFM outperforms GRAN substantially in terms of quality (VUN) on the expanded planar graph dataset (c.f. Table 4, "Stage I w/ Noise"). The same holds true for the expanded SBM dataset (c.f. Table 15, "A.16 Additional Ablations"). AFM without adversarial finetuning performs slightly worse than GRAN on the expanded lobster dataset (c.f. Table 16).
> > > - **For DiGress**: Without adversarial finetuning, AFM achieves a VUN of 23% on the planar graph dataset using 30 steps. In contrast, as shown in Figure 2a, DiGress would require at least 45 steps to achieve a VUN > 20%. Furthermore, Figure 2b indicates that in this regime, DiGress would be significantly slower during inference than AFM. Hence, even without adversarial finetuning, AFM demonstrates a superior quality-efficiency tradeoff.
> > >
> > > We agree that studying the application of adversarial finetuning to existing methods is an interesting direction for future research. We would be happy to update the manuscript to mention this as a possible avenue for future investigation. However, we believe this does not diminish the importance of developing models that inherently achieve a strong quality-efficiency tradeoff without relying on adversarial finetuning. AFM is designed to address this need.

---

### Official Review · Reviewer_thYD · 2024-11-10

**Soundness:** 2
**Presentation:** 3
**Contribution:** 2
**Rating:** 5
**Confidence:** 3

**Summary:**

The AFM utilizes graph filtration to determine a generation sequence that guides the graph generation process. In addition, a data augmentation technique based on edge perturbation, alongside an adversarial fine-tuning module, has been developed to address the problem of exposure bias associated with teacher-forcing methods. The efficiency of the proposed model compared to diffusion models has been demonstrated through numerical results.

**Strengths:**

1. The use of graph filtration in designing generation sequences for graph generation shows great promise.
2. The proposed model demonstrates significant efficiency advantages compared to diffusion models.
3. The paper is well-organized and clearly presented, making it easy for readers to understand.

**Weaknesses:**

1. The key contribution of this work lies in graph filtration and filtration sequences. While the paper clarifies how graph filtration is used to generate training samples in the form of generation sequences, and discusses the motivation behind the proposed three filtration functions and three filtration schedule sequences, it is still unclear why these filtration sequences benefit the step-by-step graph generation compared to other possible sequences. For instance, in another paper [1], the authors design a generation sequence for each graph as well. Although their task may differ slightly, as they define a pair consisting of a source graph and a target graph, they identify the "shortest" generation sequence necessary to synthesize a graph, where the concept of shortest "edit distance" justifies their generation sequences.
Additionally, if any theoretical guarantees regarding the rationale of the filtration sequences exist, they would significantly strengthen the justification for the use of graph filtration in graph generation. The results in the ablation studies section (Table 4) raise further concerns; specifically, the performance of "Stage I w/o Noise" appears to be not good, suggesting that the filtration sequences may not enhance performance. It seems that the randomness of perturbed generation sequences is what leads to improved outcomes. If I have misunderstood any aspect, please provide clarification.

2. The diversity of baselines is limited. The baselines include only two auto-regressive models, two diffusion models, and 1 GAN method. There is a lack of representation of other general graph generation models, particularly those related to normalizing flows and variational autoencoders, which restricts the completeness of the quantitative experiments. Moreover, since the authors aim to highlight the superiority of AFM over diffusion models, simply including two diffusion models as comparisons is insufficient. I recommend considering additional diffusion models (e.g., low-rank diffusion models) as competitors, as suggested in paper [2].

3. In terms of running time, AFMs only outperform diffusion models. However, the efficiency difference between AFMs and GRAN (another autoregressive model) is minimal, which negatively influences the significance of the work. Aside from running time, the experiments do not provide clear evidence that AFMs perform better than the baseline models.

References:

[1] "Graph polish: A novel graph generation paradigm for molecular optimization." IEEE Transactions on Neural Networks and Learning Systems 34.5 (2021): 2323-2337.

[2] "A survey on graph diffusion models: Generative ai in science for molecule, protein and material." arXiv preprint arXiv:2304.01565 (2023).

**Questions:**

1. I strongly recommend that the authors include a paragraph analyzing the time complexity of AFM and diffusion models in depth. Otherwise, the empirical running time alone does not convincingly support the claim of superior efficiency of AFMs.

2. Additionally, the introduction of extra hyperparameters, such as T, K, gamma(t), and lambda_t, complicates the selection of appropriate hyperparameters for a given dataset. As a solution, an extensive study or guideline for selecting these hyperparameters should be included in the manuscript’s appendix.

---

> ### Author Response · Authors · 2024-11-24
> **Author Response to Reviewer thYD (1/2)**
>
> Thank you for your time and effort in providing such constructive feedback. We greatly appreciate your insights and have addressed the points you raised below.
>
> ## Weaknesses
>
> > it is still unclear why these filtration sequences benefit the step-by-step graph generation compared to other possible sequences [...] if any theoretical guarantees regarding the rationale of the filtration sequences exist, they would significantly strengthen the justification for the use of graph filtration
>
> One key difference between our method and other autoregressive models (such as GraphRNN, GRAN, GraphGen, etc.) is that AFM can model *non-monotonic* graph sequences. While traditional methods generate graphs $G_t$ by exclusively adding edges, our model allows for both the addition and deletion of edges. This flexibility enables AFM to adjust the graph structure dynamically, correcting errors that may occur during the generation process.
>
> As we discuss in Sec. 3.2 (paragrah "Edge Decoder"), we hypothesize that the ability to both add and delete edges is crucial for avoiding the accumulation of errors during sampling (i.e., to mitigate exposure bias). If AFM samples an erroneous substructure in an intermediate step (e.g. a cycle when generating trees or a $K_5$-subdivision when generating planar graphs), it still has the opportunity to rectify this mistake in subsequent iterations.
>
> Therefore, we argue that it is essential to train AFM on noise-augmented filtrations during Stage I. By exposing the model to imperfect transitions, we explicitly train it to perform both edge additions and deletions.
>
>
> > The results in the ablation studies section (Table 4) raise further concerns; specifically, the performance of "Stage I w/o Noise" appears to be not good, suggesting that the filtration sequences may not enhance performance. It seems that the randomness of perturbed generation sequences is what leads to improved outcomes
>
> The filtration sequence is a fundamental building block of our autoregressive modeling, providing a general method to extract sequences from graphs. This filtration framework extends beyond existing autoregressive models, and we consider several natural choices for the filtration function in our work. We want to emphasize that the filtration is not a standalone component; rather, it works in conjunction with other elements, such as noise augmentation, to form our complete model. Therefore, we respectfully disagree with the hypothesis that "filtration sequences may not enhance performance."
>
> We have conducted a study on different choices of the filtration function, detailed in Appendix "A.16 Additional Ablations." Our findings indicate that some choices perform better than others, demonstrating that selecting an appropriate filtration sequence is as important as incorporating noise augmentation. This suggests that the filtration sequence plays a significant role in the overall performance of our model and optimizing the filtration function could be interesting future research direction.
>
>
> > The diversity of baselines is limited. [...] There is a lack of representation of other general graph generation models, particularly those related to normalizing flows and variational autoencoders
>
> To the best of our knowledge, there are currently no graph VAE models that can compete with the other baselines on the datasets we are considering. This observation aligns with recent related works in the field, which also do not report results for VAEs on these datasets. Moreover, we want to emphasize that running all the baselines on our datasets is time-consuming and resource-intensive; it takes up to one week to train a model on a single dataset using an A100 GPU. We have made every effort to include the most relevant methods that have demonstrated state-of-the-art performance on the small synthetic datasets from the literature.
>
> To provide a comparison to a more diverse range of baselines on the small synthetic datasets, we have included results for HiGen and EDGE to Table 1 in the revised manuscript.
>
>
> >  In terms of running time, AFMs only outperform diffusion models. However, the efficiency difference between AFMs and GRAN (another autoregressive model) is minimal
>
> Our proposed model outperforms GRAN substantially in terms of generation quality on all datasets we studied. This is consistent with results in the literature indicating that GRAN underperforms in comparison to diffusion models. While AFM is only slightly faster than GRAN on the synthetic datasets, we  note that it is 10 times faster on the larger protein graphs. **To the best of our knowledge, we present the first autoregressive graph generative model that can compete with diffuison baselines on the benchmarks we studied while being orders of magnitude faster.**

---

> ### Author Response · Authors · 2024-11-24
> **Author Response to Reviewer thYD (2/2)**
>
> ## Weaknesses
> > Aside from running time, the experiments do not provide clear evidence that AFMs perform better than the baseline models.
>
> As argued above, our experiments provide evidence that AFM substantially outperforms traditional *autoregressive* methods. We do not aim to out-perform the *diffusion* baselines in terms of quality but focus our efforts on matching their quality while substantially improving inference speed.
>
> ## Questions
> > I strongly recommend that the authors include a paragraph analyzing the time complexity of AFM and diffusion models in depth.
>
> Please refer to the "Complexity Analysis" section in our general response.
>
> > Additionally, the introduction of extra hyperparameters, such as T, K, gamma(t), and lambda_t, complicates the selection of appropriate hyperparameters for a given dataset.
>
> Thank you for this suggestion. We have added a section with practical advice on hyperparameter tuning to the appendix ("A.5 Practical Advice on Hyperparameter Choice"). Additionally, we will publish our source code upon acceptance of the paper.

---

### Official Review · Reviewer_Vm8r · 2024-11-12

**Soundness:** 2
**Presentation:** 3
**Contribution:** 2
**Rating:** 5
**Confidence:** 3

**Summary:**

The paper presents an Autoregressive Filtration Modeling (AFM) approach for graph generation, utilizing filtration to transform graphs into sequences of progressively denser subgraphs. The proposed filtration function, based on graph topology, enables a structured autoregressive generation process. This approach employs a two-stage training framework with several strategies to refine the generative model, achieving faster generation than existing autoregressive models.

**Strengths:**

1. The presentation is organized.
2. The paper introduces noise augmentation and reinforcement learning approach, reducing the model's exposure bias, which is common in autoregressive models.
3. The ablation studies are comprehensive. The ablations reveal the importance of noise augmentation and adversarial fine-tuning and demonstrate the effect of the different filtration function.

**Weaknesses:**

1. A comparative analysis with other approaches, such as the autoregressive model by Kong et al. (2023) and the EDGE graph diffusion model by Chen et al. (2023), would be helpful.
2. An analysis on the impact of filtration steps is missing; it would be insightful to see how generation quality changes as the number of filtration steps varies.
3. Another potential limitation, as acknowledged by the authors, is that the current filtration function design primarily considers the graph’s topology, which may not fully satisfy structural consistency for attributed graphs. Exploring ways to incorporate attribute information could be beneficial.

**Questions:**

Q1: Have you explored how varying the number of filtration steps might impact generation quality, especially on larger and more complex graphs? It would be helpful to understand if fewer filtration steps lead to notable quality degradation or if additional steps have limited benefit.

Q2: Given that the experiments were conducted on graphs with up to 500 nodes, have you considered evaluating the model’s performance on larger, real-world datasets? It would be interesting to see how well the approach performs on larger graphs, such as Cora, Citeseer, or Polblogs (with over 1000 nodes). It could provide valuable insights into the model’s feasibility for high-throughput, large-scale applications.

---

> ### Author Response · Authors · 2024-11-24
> **Author Response to Reviewer Vm8r (1/2)**
>
> Thank you for your time and effort in providing such constructive feedback. We greatly appreciate your insights and have addressed the points you raised below.
>
> ## Weaknesses
>
> > A comparative analysis with other approaches, such as the autoregressive model by Kong et al. (2023) and the EDGE graph diffusion model by Chen et al. (2023), would be helpful.
>
> We have added a detailed discussion of the similarities and differences of absorbing state diffusion (particularly GraphARM and EDGE) and AFM to our extended related work section in the appendix (A.1).
>
> Additionally, we have added results of EDGE to table 1. Unfortunately, Kong et al. have not released official code for GraphARM.
>
> > An analysis on the impact of filtration steps is missing; it would be insightful to see how generation quality changes as the number of filtration steps varies.
>
> As noted in the common response, we have added this experiment to the main text.
>
> > Exploring ways to incorporate attribute information could be beneficial.
>
> We acknowledge the concern that our current design does not explicitly support graphs with node and edge attributes. However, we want to emphasize that our primary focus is on developing efficient AR model for unattributed graphs and we present the **first AR model that matches the performance of diffusion-based models while maintaining its speed advantage**. Furthermore, un-attributed graph topologies are often of significant interest in various applications, such as protein structure modeling and design, where contact maps represent the spatial proximity between amino acids [1, 2].
>
> As discussed in our conclusion and appendix, our method is naturally extendable to edge-labeled graphs. Regarding the modeling of node attributes, we propose two prospective approaches:
> - Post-hoc node labeling: After generating the graph topology, node labels can be assigned based on the structural context, as suggested by Bergmeister et al. (2024). For example, in molecular graphs, valency constraints and local connectivity patterns may provide strong cues for determining atom types. Such an approach was used in [3].
> - Extension of the Filtration Process: We can extend the filtration process to include node labels by representing them as labeled self-loops and incorporating them into the edge filtration. This approach allows the model to learn node attributes alongside the graph topology within the same generative framework.
>
> We believe that integrating node and edge labels is a promising direction for future work, which can further enhance the applicability of our approach to a broader range of graph generation tasks.

---

> ### Author Response · Authors · 2024-11-24
> **Author Response to Reviewer Vm8r (2/2)**
>
> ## Questions
> > Have you explored how varying the number of filtration steps might impact generation quality
>
> As noted in the common response, we have added this experiment to the main text.
>
> > It would be interesting to see how well the approach performs on larger graphs, such as Cora, Citeseer, or Polblogs
>
> We would like to clarify that some of the protein graphs used in our experiments are already moderately large, containing up to 500 nodes.
> Currently, we are not aware of publicly available datasets with a reasonable sample size that would allow us to properly benchmark our method on graphs consisting of more than 1K nodes.
>
> - Point Cloud dataset by Liao et al. (2020): This dataset includes graphs with up to 5,000 nodes but contains only 26 graphs in the training set and 8 graphs in the test set. Training a generative model and performing a thorough evaluation on such a small dataset is challenging due to high variance and limited statistical significance.
> - Single Large Graphs (e.g., EDGE with CiteSeer or Cora): Some works have trained models on individual large graphs. However, this approach focuses on generating a single graph rather than learning a distribution over graphs, which is the primary objective of our work.
> - Subgraphs from Larger Networks (e.g., GraphARM): Other approaches sample subgraphs from larger citation networks. Unfortunately, these subgraphs are generally small, with sizes up to 87 nodes in the case of the Cora dataset used by GraphARM. This does not address the need for experiments on significantly larger graphs.
>
> If there are specific datasets that you can recommend for evaluating generative models on larger graphs, we would be eager to consider them. However, we also want to clarify that our goal is to demonstrate the **good balance between efficiency and generation quality** of our method in high-throughput applications, rather than the scalability to generating larger graphs. To clarify this point, we have revised our manuscript to replace the term "scalability" with the more precise term "efficiency" and "high-throughput generation" (see lines 24, 64, 74 of original submission).
>
> ## References
> [1] Zeming Lin et al., Evolutionary-scale prediction of atomic-level protein structure with a language model. Science379, 1123-1130(2023). DOI:10.1126/science.ade2574
>
> [2] Chen, Y., Xu, Y., Liu, D. et al. An end-to-end framework for the prediction of protein structure and fitness from single sequence. Nat Commun 15, 7400 (2024). https://doi.org/10.1038/s41467-024-51776-x
>
> [3] Li, Yibo, Jianxing Hu, Yanxing Wang, Jielong Zhou, Liangren Zhang, and Zhenming Liu. ‘DeepScaffold: A Comprehensive Tool for Scaffold-Based De Novo Drug Discovery Using Deep Learning’. Journal of Chemical Information and Modeling 60, no. 1 (2020): 77–91. https://doi.org/10.1021/acs.jcim.9b00727.

---

### Author Response · Authors · 2024-11-24
**General Author Response**

We sincerely thank all the reviewers for their time and great efforts in providing helpful and detailed feedback, which we felt has strengthened our paper. We have incorporated several important changes into the revised PDF, highlighted in red.

We appreciate the reviewers recognizing several strengths of our work: Well-written and organized presentation (Vm8r, thYD, GgH1, xUTE), technical novelty (Vm8r, vHcj, JXGh), and technical merits (Vm8r, 7YaU, GgH1, vHcj, JXGh).

Below, we address some common points that were raised by multiple reviewers.

## Scope of our work

Our primary goal is to advance the field of graph generation by offering a method that balances performance and efficiency, particularly for high-throughput applications. We concentrate on unattributed graphs because they are fundamental in many domains and serve as a cornerstone toward more complex attributed graph generation. To the best of our knowledge, we propose the first autoregressive model that achieves performance comparable to diffusion models while being orders of magnitude faster during inference. By demonstrating substantial speedups without compromising quality on graphs of practical sizes, we believe our work makes a significant contribution to the community.


## Complexity analysis
- **Asymptotic complexity of sampling**: We have provided a comprehensive analysis of the computational complexity of our proposed method to address the reviewers' concerns. Sampling a graph with $N$ nodes from an AFM with $T$ generation steps has an asymptotic complexity of $\mathcal{O}(T^2N + TN^3)$, where the cubic dependence on $N$ arises from the computation of Laplacian positional embeddings. The practical efficiency of our method is nevertheless mainly due to our ability to use a small number of timesteps $T$ (30 or 15 in our settings). Notably, such a small number of steps is not feasible for diffusion-based models to achieve comparable performance (see Figure 2 in our paper). We have briefly stated the asymptotic complexity for sampling from AFM in the main text and provided a detailed derivation in the appendix ("A.2 Complexity Analysis"), including comparisons to baseline methods.
- **Asymptotic dependence on $T$**: Two reviewers noted that the runtime and memory complexities of our method are quadratic and linear in $T$, respectively, due to our temporal mixing operations. They were concerned that this might lead to issues when $T$ is large. In our revised revision, we have investigated a variant of AFM that employs a first-order autoregressive structure, reducing the time complexity to linear in $T$ and making the memory complexity independent of $T$. In practice, we find that this first-order variant is not substantially faster and we do not encounter memory bottlenecks with our original method. However, the first-order structure could serve as a viable alternative if a very large $T$ is desired. Please refer to Section "A.3 A First-Order Autoregressive Variant" for more details.




## Effect of the filtration granularity $T$
The efficiency of our method is primarily driven by our ability to generate graphs with very few generation steps, controlled by the parameter $T$. Reviewer Vm8r and GgH1 inquired about the quality-efficiency trade-off determined by varying $T$, while reviewer vHCj wondered whether better trade-offs could be achieved in diffusion models (specifically DiGress) by performing fewer denoising steps.

To address these concerns, we conducted experiments detailed in the main text (see Section 4.4 "Ablation Studies"), where we trained both AFM and DiGress models with different values of $T$ on the expanded planar graph dataset. Our findings are as follows:

- The quality of AFM-generated samples begins to decrease when $T$ is set too large. This suggests that our method is optimized for a smaller number of generation steps
- DiGress suffers significantly when $T$ is chosen to be small. The model requires a larger number of denoising steps to maintain sample quality, indicating less flexibility in trading off between efficiency and quality.
- DiGress is generally slower than AFM, regardless of the choice of $T$. This consistent difference highlights the efficiency advantage of our method over diffusion-based models like DiGress.

---

### Author Response · Authors · 2024-11-29
**Kind Reminder to Reviewers**

As the discussion period is coming to a close, we kindly remind the reviewers who have not yet responded to our rebuttal to consider our responses and updated manuscript. Please let us know if you have any additional questions or doubts, we are happy to continue the discussion!

---

### Meta-Review · Area_Chair_PFYG · 2024-12-20

**Metareview:**

This paper address the graph generation problem by introducing Autoregressive Filtration Modeling (AFM) to improve computational efficiency for learning and generating graph samples while keeping high graph quality generated. The quality of generated graphs are measured in the sense of MMD metric. While there is (incremental) improvement in terms of versatile network statistics and metric for the presented settings, as reviewers pointed out, the computational time improvement discussed is less clear, weakening the computational gain part of the claim. Moreover, the autoregressive structure using filtration function is less clearly explained, that veils . As reviewers suggest, additional discussion on the comparison and efficiency may need additional discussions and details, including those discussed during the revision period. Given the interesting idea and existing insights explored, we strongly recommend another round of revision and polish to appear in major conference such as ICLR.

**Additional Comments On Reviewer Discussion:**

The reviewers' concerns are partially addressed.

---

### Decision · Program_Chairs · 2025-01-22

Reject